# DNAChunker: Learnable Tokenization for DNA Language Models

**Taewon Kim** [1]   **Jihwan Shin** [2]   **Hyomin Kim** [1]   **Youngmok Jung** [3]   **Jonghoon Lee** [3]   **Won-Chul Lee** [3]
**Sungsoo Ahn** [1] [†]   **Insu Han** [2] [†]

## Abstract

DNA language models are increasingly used to represent genomic sequence, yet their effectiveness depends critically on how raw nucleotides are converted into model inputs. Unlike natural language, DNA offers no canonical boundaries, making fixed tokenizations a brittle design choice under shifts, indels, and local repeats. We introduce DNACHUNKER, a masked DNA language model that incorporates a learnable adaptive segmentation module to produce context-dependent, variable-length units. Building on a dynamic segmentation procedure, DNACHUNKER learns to allocate finer granularity to functionally enriched regions while compressing repetitive or redundant sequence. We pretrain DNACHUNKER on the human reference genome and evaluate it across five benchmarks, where it consistently improves over strong fixed-tokenization baselines. Further analyses and ablations indicate that unlike fixed tokenizations, segmentation is learned in a biologically-informed, mutation-resilient manner.

## 1. Introduction

DNA sequences encode the regulatory and molecular programs that underlie life, from gene regulation (Moore et al., 2020; Kellis et al., 2014) and protein synthesis (Jia et al., 2024) to DNA replication (Ekundayo & Bleichert, 2019). Rapid progress in sequencing technologies (Behjati & Tarpey, 2013) has transformed genomics into a data-rich field, producing sequence data at unprecedented scale (Chen et al., 2021). Yet, accurately modeling the functions specified by these sequences remains a central challenge (Libbrecht & Noble, 2015; Li et al., 2023). Genomes are exceptionally long, function is often context-dependent, and

†Indicates co-corresponding authors. [1]Graduate School of AI, KAIST [2]School of Electrical Engineering, KAIST [3]Inocras Korea Inc. Correspondence to: Sungsoo Ahn <sungsoo.ahn@kaist.ac.kr>, Insu Han <insu.han@kaist.ac.kr>.

*Proceedings of the $43^{rd}$ International Conference on Machine Learning*, Seoul, South Korea. PMLR 306, 2026. Copyright 2026 by the author(s).

high-quality annotated datasets remain limited, making it difficult to infer the principles that govern biological sequence function (Kellis et al., 2014; Libbrecht & Noble, 2015).

Inspired by the success of large language models (LLMs; Anil et al., 2023), several recent works have begun investigating DNA language models (Ji et al., 2021; Sanabria et al., 2024; Dalla-Torre et al., 2025), moving beyond traditional rule-based methods to learn the "grammar" and "semantics" of DNA. In particular, the presence of long-range interactions between nucleotides and functional elements such as promoters and enhancers that act as "words" in the genomic language highlights the need for a tokenization strategy that can group DNA sequences into meaningful tokens.

Genomic sequences pose unique challenges for tokenization that differ from natural language, primarily due to the absence of a natural "word" unit. Prior works have largely adopted one of three approaches: single nucleotides (Dalla-Torre et al., 2025; Schiff et al., 2024), fixed-size k-mers (Poli et al., 2023; Ji et al., 2021), or Byte-Pair Encoding (BPE; Zhou et al., 2024; Sanabria et al., 2024). The single nucleotide approach, while simple, results in excessively long sequences that make it computationally expensive and difficult to model long-range interactions (Dalla-Torre et al., 2025).

To circumvent this length issue, fixed-size k-mers and BPE have been explored, but these methods are inherently fixed and struggle to adapt to the biological context of DNA. As demonstrated in Figure 1b, k-mer tokenization is highly sensitive to small shifts, where a single insertion, deletion, or mutation can completely alter the tokenized output, even if the biological function remains unchanged (Dalla-Torre et al., 2025). Next, frequency-driven schemes like BPE fail to capture the functional importance of substrings, since the most frequent substrings are typically simple non-functional repetitive elements. Figure 1c visualizes this effect: *both* k-mer tokenization and BPE actively fragment known genomic motifs such as TF-binding and cis-regulatory motifs.

To this end, we propose DNACHUNKER, a bidirectional DNA language model trained with masked language modeling, designed to overcome the limitations of fixed tokenization. Our model integrates a learnable, dynamic tok-

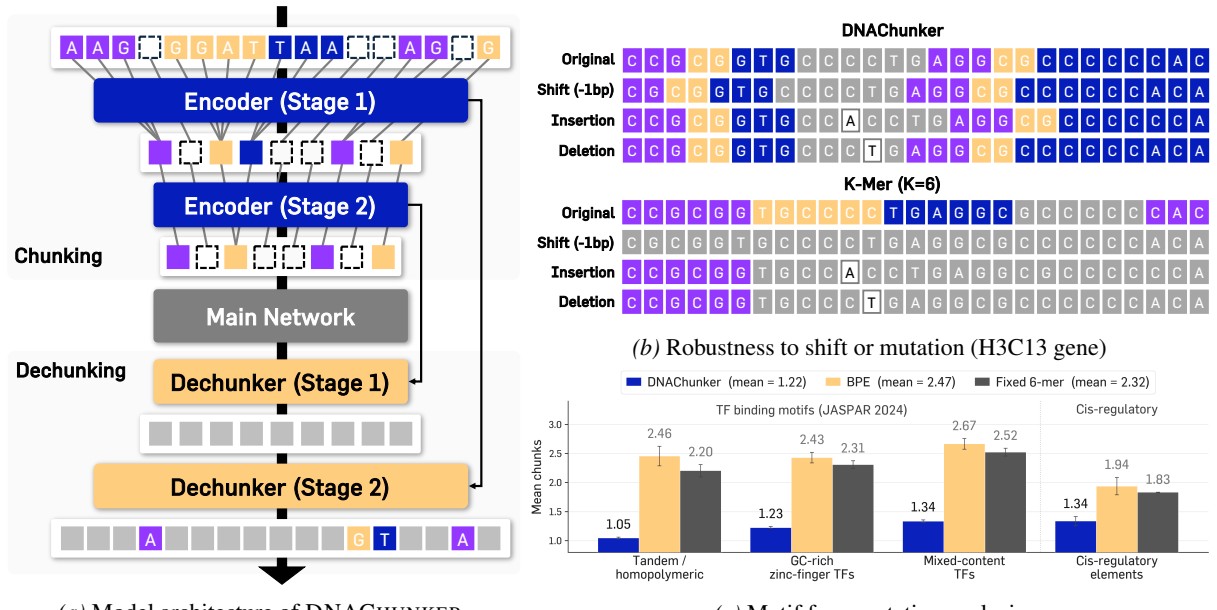

*(a)* Model architecture of DNACHUNKER

*(b)* Robustness to shift or mutation (H3C13 gene)

*(c)* Motif fragmentation analysis

*Figure 1.* **Architecture, tokenizer robustness, and motif fragmentation analysis.** (a) The architecture of DNACHUNKER. (b) Our tokenizer is robust against nucleotide-wise shifts or mutations, where we color the tokens to indicate that they are preserved despite the mutations. (c) DNACHUNKER dynamically allocates tokens based upon input context, and thus unlike BPE, DNACHUNKER recognizes motifs as a singular functional chunk.

enizer that segments DNA into variable-length, context-dependent chunks, producing biologically meaningful groupings learned directly from genomic sequence.

Concretely, we use a two-stage hierarchical encoder: raw base-pair embeddings are first processed with lightweight bidirectional Mamba layers (Schiff et al., 2024), then adjacent positions are merged into chunks using cosine-similarity–based boundary decisions, grouping highly similar representations into a single token. The resulting compressed sequence is modeled by a more expressive main network to capture long-range dependencies, and is then upsampled back to base-pair resolution via a bidirectional dechunking module that reconstructs fine-grained representations while leveraging residual connections from encoder features. This enables efficient long-context modeling while retaining nucleotide-level fidelity for masked prediction.

We validate DNACHUNKER by pretraining on the human reference genome (GRCh38/hg38) and evaluating it on *five benchmarks*: the Nucleotide Transformer benchmark and its revised version (NT benchmark; Dalla-Torre et al., 2025), Genomic Benchmarks (Grešová et al., 2023), BEND (Marin et al., 2024), and DNALongBench (Cheng et al., 2025). Together, these benchmarks span a broad spectrum of genomic prediction problems, from short-range regulatory and splice-site annotation to epigenomic profiling, variant effect prediction, and long-range regulatory interactions over contexts up to megabase scale. Across this diverse evaluation suite, DNACHUNKER demonstrates strong performance, outperforming larger baselines that rely on multi-species

pretraining (Dalla-Torre et al., 2025; Wu et al., 2025).

In addition to downstream accuracy, we show that the learned tokenizer produces biologically meaningful and mutation-robust segmentations, preserving functional motifs and maintaining stable boundaries under diverse variants. Controlled ablations confirm that each proposed component contributes to performance, while FLOPs analyses show that adaptive segmentation improves long-context efficiency by reducing the effective sequence length. Together, these results suggest that DNACHUNKER offers a more effective and efficient approach to genomic sequence modeling.

Overall, our contributions are summarized as follows:

- **Bidirectional dynamic tokenization.** We adapt dynamic tokenization from autoregressive language modeling to the masked DNA pretraining setting, enabling bidirectional modeling of token boundaries (Section 3.1).

- **Strong performance across benchmarks.** DNACHUNKER achieves state-of-the-art or competitive performance across five benchmarks spanning regulatory, epigenomic, variant-effect, and long-range tasks — using only GRCh38/hg38 for pretraining (Section 4.1).

- **Meaningful and robust genomic tokens.** The learned tokenizer preserves functional genomic motifs as singular chunks, remains stable under SNVs, InDels, and structural variants, and improves long-context efficiency by reducing the effective sequence length (Section 4.2).

**Conflict of Interest Disclosure.** Authors Y.J., J.L. and W.-C.L. are affiliated with Inocras Korea Inc., which contributed to the development of DNACHUNKER. The proposed model was developed in collaboration with Inocras.

## 2. Related Works

### 2.1. DNA Language Models

**Autoregressive DNA models.** Generative DNA modeling has rapidly progressed from early next-token prediction approaches like DNAGPT (Zhang et al., 2024) to long-context architectures such as HyenaDNA (Nguyen et al., 2023) and megaDNA (Shao & Yan, 2024), which leverage Hyena (Poli et al., 2023) and multiscale transformers to extend sequence length. Scaling this paradigm, Evo (Nguyen et al., 2024) and Evo2 (Brixi et al., 2026) train on trillions of base pairs across prokaryotic and viral genomes, enabling real-world applications such as CRISPR-Cas (Nguyen et al., 2024) and bacteriophage design (King et al., 2025). Despite these advances, autoregressive models are inherently unidirectional and thus produce suboptimal representations, for genomic signals depend on both upstream and downstream context.

**Masked DNA models.** Masked language models (MLMs) better reflect the bidirectional nature of DNA, where regulatory elements act in both directions and functional prediction requires upstream and downstream context. As a result, autoregressive DNA models typically require substantially more parameters or data to match MLM accuracy on predictive tasks (Shu et al., 2026). The Nucleotide Transformer (NT; Dalla-Torre et al., 2025) scales MLM pretraining to 2.5B parameters across multispecies genomes, while DNABERT-2 (Zhou et al., 2024) and GROVER (Sanabria et al., 2024) adopt BPE tokenization over k-mers. To overcome the context-length bottleneck of standard transformers, GENA-LM (Fishman et al., 2025) uses sparse attention and Caduceus (Schiff et al., 2024) adopts BiMamba (Tang et al., 2024). However, all of these rely on fixed tokenization schemes that are blind to genomic context.

### 2.2. Learnable Tokenizers

Learnable tokenization has primarily been studied in the autoregressive setting. Delimiter-based methods like Space-Byte (Slagle, 2024) and entropy-based approaches such as the Byte Latent Transformer (Pagnoni et al., 2025) identify boundaries through external signals, while H-Net (Hwang et al., 2026) learns dynamic chunking end-to-end and matches fixed-tokenizer baselines on language and DNA. For non-autoregressive models, prior work has explored gradient-based multi-resolution pooling (Tay et al., 2022) and word-level external chunking (Thawani et al., 2023; Sreedhar et al., 2023), but these still depend on predefined boundary heuristics rather than learning segmentation

jointly with the representation. DNACHUNKER is, to our knowledge, the first to integrate fully learnable, dynamic chunking into a masked DNA language model, jointly learning bidirectional representations and biologically meaningful token boundaries.

## 3. Methodology

### 3.1. Architecture Details of DNACHUNKER

DNACHUNKER is a bidirectional masked language model (MLM) for genomic sequences, structured around a central design principle: the main network serves as the primary locus of long-range contextual reasoning. Operating on adaptively compressed segment sequences, the main network aggregates evidence across distant genomic regions while remaining computationally tractable. The encoder and decoder are designed to support this goal—compressing sequences into the main network and reconstructing base-pair predictions from it—while addressing two key challenges.

First, genomic signals depend on both upstream and downstream context, demanding bidirectionality throughout. Most prior adaptive segmentation methods were introduced in autoregressive settings (Pagnoni et al., 2025; Hwang et al., 2026), making them inherently directional. In contrast, DNACHUNKER employs bidirectional Mamba layers to inform segmentation decisions from both flanks in the encoder, while the decoder applies bidirectional probability-gated smoothing during reconstruction.

Second, `[MASK]` tokens are functional artifacts of pretraining, not semantic nucleotides, and require careful handling to prevent information leakage. In the encoder, we enforce boundaries around masked positions so that segmentation decisions remain mask-invariant and transfer to downstream sequences without masks. In the decoder, we gate residual connections to masked positions, ensuring their reconstruction depends solely on main-network representations rather than leaked encoder context.

Together, these designs concentrate contextual modeling within the main network while enabling adaptive compression that respects the bidirectional and mask-sensitive nature of MLM pretraining. We illustrate the architecture in Figure 1a and provide hyperparameters in Section A. Below, we describe the encoder (Section 3.1.1), main network (Section 3.1.2), and decoder (Section 3.1.3) in detail. We summarize our pretraining and finetuning details in Section B and Section C.

#### 3.1.1. ENCODER: MASK-PROTECTED ADAPTIVE SEGMENTATION

The encoder compresses genomic sequences via *adaptive segmentation*, reducing the effective sequence length in low-

information regions while preserving high-information content at an appropriate granularity. DNACHUNKER applies a two-stage hierarchical procedure that progressively maps base pair–level signals into coarser, semantically meaningful representations. Each stage follows three steps: (1) encode the current sequence into contextualized embeddings, (2) infer decision boundaries between adjacent positions, and (3) downsample embeddings according to these boundaries to produce a shorter stage output. This structured workflow retains salient genomic patterns while reducing computational cost.

In practice, step (1) is handled by a lightweight *bidirectional* Mamba backbone (Schiff et al., 2024), which transforms raw tokens into base-pair–resolution features optimized for boundary inference. Steps (2)–(3) are performed by the segmentation module, which predicts boundaries from these processed features and aggregates embeddings to form segment representations. Bidirectionality is important for genomic sequences because boundary evidence can arise from both upstream and downstream context, unlike autoregressive settings that typically constrain encoders to a single direction (Pagnoni et al., 2025; Hwang et al., 2026).

Formally, given an input sequence of length $T$, let $x^{(0)} = (x_1^{(0)}, \ldots, x_T^{(0)})$ denote base pair–level embeddings. These embeddings are processed by the first-stage encoder, producing intermediate representations $\widehat{x}^{(0)}$. A routing network then computes boundary probabilities $p_t^{(0)}$ for each position $t \in [1, T]$ using cosine dissimilarity between projected query and key vectors:

$$p_t^{(0)} = \frac{1}{2}\left(1 - \frac{(q_t^{(0)})^\top k_{t-1}^{(0)}}{\|q_t^{(0)}\| \cdot \|k_{t-1}^{(0)}\|}\right),$$
$$q_t^{(0)} = W_{\text{enc},q}^{(1)} \widehat{x}_t^{(0)}, \qquad k_t^{(0)} = W_{\text{enc},k}^{(1)} \widehat{x}_t^{(0)}, \tag{1}$$

where $W_{\text{enc},q}^{(s)}$ and $W_{\text{enc},k}^{(s)}$ are learnable parameters of the encoder routing network at stage $s \in \{1, 2\}$. We obtain hard boundary indicators by thresholding:

$$b_t^{(s)} = \mathbf{1}\left(p_t^{(s)} \geq 0.5\right). \tag{2}$$

These indicators define chunk boundaries. The first stage collects $T' = \sum_{t=1}^{T} b_t^{(0)}$ adaptive chunks from $\widehat{x}^{(0)}$, yielding chunked embeddings $x^{(1)} \in \mathbb{R}^{T' \times d}$. The second-stage encoder applies the same procedure to $x^{(1)}$ to produce a more coarse-grained representation $x^{(2)} = (x_1^{(2)}, \ldots, x_{T''}^{(2)})$ with $T'' < T'$, which is used as input to the main network.

**Mask protection mechanism.** A key complication in MLM pretraining is that [MASK] is a *functional* token rather than a semantic nucleotide. Since the routing network predicts boundaries from contextualized features, allowing [MASK] to merge with neighboring bases can introduce

shortcuts: the model may learn segmentation patterns that exploit mask placement (a pretraining-only artifact) rather than genomic context, which does not transfer to downstream sequences without masked tokens. To prevent mask-conditioned tokenization, we enforce boundaries around every masked base pair so that masked positions are never merged into larger chunks.

Concretely, for each masked index $m$, we force chunk boundaries immediately before and after it, ensuring that the masked token forms a singleton chunk and its neighbors start new chunks. This mask protection is applied throughout the hierarchical encoder, ensuring that masked tokens remain isolated at every segmentation stage. As a result, adaptive segmentation decisions are driven by genomic context rather than by the presence of [MASK], while the MLM learning signal is preserved through compression.

### 3.1.2. MAIN NETWORK

The main network consists of 30 Transformer blocks operating on the compressed segment sequence. Each block follows the standard Transformer design with layer normalization, multi-headed self-attention, and a feedforward network with GELU (Hendrycks & Gimpel, 2016) activation. We incorporate Rotary Position Embeddings (RoPE; Su et al., 2024) in the attention mechanism to encode positional information, utilizing the mean index location of each chunk. The main network accounts for the majority of parameters in DNACHUNKER and memory usage during inference. The main network is intended to serve as the primary locus for contextual modeling of the input DNA sequence.

### 3.1.3. DECODER: HIERARCHICAL DECHUNKING WITH BIDIRECTIONAL SMOOTHING

Mirroring the encoder's two-stage adaptive segmentation, the decoder reconstructs representations in two hierarchical steps ($z^{(0)} \to z^{(1)} \to z^{(2)}$), progressively expanding the compressed sequence back to base-pair resolution. Unlike autoregressive reconstructions (Hwang et al., 2026; Pagnoni et al., 2025) that must remain causal, our bidirectional MLM decoder leverages context from both directions.

Each dechunking step proceeds as follows. Given compressed representations $z^{(s)} \in \mathbb{R}^{T_s \times d}$ and boundary indicators $b^{(S-s)}$ from the corresponding encoder stage, we first *paste* each segment representation to all positions it governs via cumulative boundary counts:

$$\tilde{z}_t^{(s+1)} = z_{\sum_{k=1}^{t} b_k^{(S-s)}}^{(s)}. \tag{3}$$

Here, $S$ denotes the total number of hierarchical layers. This initial assignment produces piecewise-constant representations. To (1) enable gradient flow through discrete boundary

decisions and (2) incorporate bidirectional context, we then apply probability-gated smoothing in *both directions*:

$$z_t^{(s+1)} = \tfrac{1}{2}\big(\text{SCAN}_{\rightarrow}(\tilde{z}^{(s+1)}, p)_t + \text{SCAN}_{\leftarrow}(\tilde{z}^{(s+1)}, p)_t\big),$$
(4)

where $\text{SCAN}_{\rightarrow}$ and $\text{SCAN}_{\leftarrow}$ denote forward and backward linear recurrences gated by boundary probabilities $p$. The smoothed representations are then combined with gated encoder residuals from the corresponding stage and refined by bidirectional Mamba layers, before proceeding to the next dechunking step or the final language model head.

**Masked residual gating.** At each dechunking stage, the decoder employs residual connections from the corresponding encoder features to aid reconstruction of fine-grained information. However, allowing these residuals to flow into masked positions creates an undesirable shortcut: since the encoder's bidirectional layers mix information across neighbors, masked tokens could be reconstructed from leaked encoder context alone, bypassing the main network. To enforce contextual compute in the main network, we gate residual connections based on whether a position's assigned segment contains a mask token—positions in masked segments receive zero residual. This design ensures that the main network serves as the primary locus of long-range dependency modeling.

### 3.2. Model Pretraining

**Loss function.** DNACHUNKER is pretrained with masked language modeling, with down-weighting of repetitive regions of DNA by 0.1, in line with prior works (Brixi et al., 2026). The loss is formulated as follows:

$$\mathcal{L}_{\text{MLM}} = \sum_{t \in M} w_t\, \mathcal{L}_{\text{CE}}(t),$$
$$w_t = \begin{cases} 0.1 & \text{if position } t \text{ is in a repetitive region,} \\ 1.0 & \text{otherwise.} \end{cases}$$
(5)

where $\mathcal{L}_{\text{CE}}(t)$ denotes the cross entropy loss for predicting the masked nucleotide at position $t$. Additionally, to control the degree of compression from the chunking layers, we use the ratio loss proposed by Hwang et al. (2026):

$$\mathcal{L}_{\text{ratio}}^{(s)} = \frac{\bar{b}^{(s)}\,\bar{p}^{(s)}}{\alpha^{(s)}} + \frac{\big(1 - \bar{b}^{(s)}\big)\big(1 - \bar{p}^{(s)}\big)}{1 - \alpha^{(s)}},$$
$$\bar{b}^{(s)} = \frac{1}{T}\sum_{t=1}^{T} b_t^{(s)}, \qquad \bar{p}^{(s)} = \frac{1}{T}\sum_{t=1}^{T} p_t^{(s)}.$$
(6)

where $\bar{b}^{(s)}$ and $\bar{p}^{(s)}$ are the fraction of selected tokens and the average boundary probability, respectively, and $\alpha^{(s)} \in (0,1)$ is the target compression ratio of the encoder, which is a controllable parameter. Note that $\bar{b}^{(s)}$ is non-differentiable,

but the network can be trained towards the target compression ratio through tuning $\bar{p}^{(s)}$. Together, we train the model to minimize the loss $\mathcal{L} = \mathcal{L}_{\text{MLM}} + \lambda \mathcal{L}_{\text{ratio}}^{(0)} + \lambda \mathcal{L}_{\text{ratio}}^{(1)}$, where $\lambda$ is the weighting coefficient. More details about pretraining can be found in Section B.

**Dataset.** We pretrain our model on the Human Reference Genome, adopting the data partitioning strategy from Enformer (Avsec et al., 2021). The genome is first divided into non-overlapping regions of $2^{20}$ (1,048,576) base pairs (bp), which will be allocated to the training, validation, and test sets. These regions are subsequently segmented into input sequences with a maximum length of 8192 bp. During the preprocessing, ambiguous nucleotides ('N') are mapped to a padding token and are excluded from the loss computation. Following the methodology of BERT (Devlin et al., 2019), for each input sequence, 15% of all nucleotides are randomly selected for prediction. Of this selection, 80% are replaced with a [MASK] token, 10% are substituted with a random nucleotide, and the remaining 10% are left unchanged.

**Fine-tuning on downstream tasks.** For fine-tuning on the downstream tasks, we remove the language model head and perform average pooling over the valid tokens, *i.e.* excluding [PAD] tokens. The pooled output is subsequently passed through a linear layer. Depending upon the dataset and evaluation protocol, we perform full-finetuning or linear probing (*i.e.* freezing the main model while tuning only the linear head). Details are provided in Section C.

## 4. Experiments

We evaluate DNACHUNKER on five benchmarks spanning short- and long-range genomic tasks: the Nucleotide Transformer benchmark (NT benchmark) and its revised version (Dalla-Torre et al., 2025), the Genomic Benchmark (Grešová et al., 2023), BEND (Marin et al., 2024), and DNALongBench (Cheng et al., 2025). Despite using only 172M parameters, DNACHUNKER achieves strong performance across all five benchmarks (Section 4.1). We then conduct ablative studies (Section 4.2) that (i) compare against prior DNA-targeted tokenization schemes, (ii) isolate the contribution of each architectural component, (iii) quantify computational overhead, and (iv) probe the biological structure captured by the learned tokenizer.

### 4.1. Downstream Tasks

**Nucleotide Transformer benchmark.** We evaluate our model on the NT benchmark in Table 1, where DNACHUNKER achieves state-of-the-art performance on 13 out of 18 datasets and the best total average MCC (0.772) and average rank (1.67), improving over the next-best baseline

*Table 1.* **Nucleotide Transformer Benchmark.** Performance on the NT benchmark. Values report Matthews correlation coefficient (MCC; mean $\pm$ standard error) over 10-fold cross-validation. Higher is better for MCC; lower is better for average rank. Best results are **bold**; second-best results are underlined.

| | Enformer (252M) | DNABERT-2 (117M) | HyenaDNA (55M) | NT-multi (2.5B) | NT-v2 (500M) | Caduceus-PH (8M) | Caduceus-PS (8M) | GROVER (87M) | GENERator (1.2B) | DNACHUNKER (172M) |
|---|---|---|---|---|---|---|---|---|---|---|
| *Histone Markers* | | | | | | | | | | |
| H3 | $0.724 \pm 0.018$ | $0.785 \pm 0.012$ | $0.781 \pm 0.015$ | $0.793 \pm 0.013$ | $0.788 \pm 0.010$ | $0.794 \pm 0.012$ | $0.772 \pm 0.022$ | $0.768 \pm 0.008$ | $\underline{0.806} \pm 0.005$ | $\mathbf{0.817} \pm 0.011$ |
| H3K14ac | $0.284 \pm 0.024$ | $0.515 \pm 0.009$ | $\underline{0.608} \pm 0.020$ | $0.538 \pm 0.009$ | $0.538 \pm 0.015$ | $0.564 \pm 0.033$ | $0.596 \pm 0.038$ | $0.548 \pm 0.020$ | $0.605 \pm 0.008$ | $\mathbf{0.711} \pm 0.021$ |
| H3K36me3 | $0.345 \pm 0.019$ | $0.591 \pm 0.005$ | $0.614 \pm 0.014$ | $0.618 \pm 0.011$ | $0.618 \pm 0.015$ | $0.590 \pm 0.018$ | $0.611 \pm 0.048$ | $0.563 \pm 0.017$ | $\underline{0.657} \pm 0.007$ | $\mathbf{0.677} \pm 0.003$ |
| H3K4me1 | $0.291 \pm 0.016$ | $0.512 \pm 0.008$ | $0.512 \pm 0.008$ | $0.541 \pm 0.009$ | $0.544 \pm 0.009$ | $0.468 \pm 0.015$ | $0.487 \pm 0.029$ | $0.461 \pm 0.018$ | $\underline{0.553} \pm 0.009$ | $\mathbf{0.631} \pm 0.009$ |
| H3K4me2 | $0.207 \pm 0.021$ | $0.333 \pm 0.013$ | $\underline{0.455} \pm 0.028$ | $0.324 \pm 0.014$ | $0.302 \pm 0.020$ | $0.332 \pm 0.034$ | $0.431 \pm 0.016$ | $0.403 \pm 0.042$ | $0.424 \pm 0.013$ | $\mathbf{0.599} \pm 0.021$ |
| H3K4me3 | $0.156 \pm 0.022$ | $0.353 \pm 0.021$ | $\underline{0.550} \pm 0.015$ | $0.408 \pm 0.011$ | $0.437 \pm 0.028$ | $0.490 \pm 0.042$ | $0.528 \pm 0.033$ | $0.458 \pm 0.022$ | $0.512 \pm 0.009$ | $\mathbf{0.660} \pm 0.045$ |
| H3K79me3 | $0.498 \pm 0.013$ | $0.615 \pm 0.010$ | $0.669 \pm 0.014$ | $0.623 \pm 0.010$ | $0.621 \pm 0.012$ | $0.641 \pm 0.028$ | $\underline{0.682} \pm 0.018$ | $0.626 \pm 0.026$ | $0.670 \pm 0.011$ | $\mathbf{0.731} \pm 0.012$ |
| H3K9ac | $0.415 \pm 0.020$ | $0.545 \pm 0.009$ | $0.586 \pm 0.021$ | $0.547 \pm 0.011$ | $0.567 \pm 0.020$ | $0.575 \pm 0.024$ | $0.564 \pm 0.018$ | $0.581 \pm 0.015$ | $\underline{0.612} \pm 0.006$ | $\mathbf{0.678} \pm 0.007$ |
| H4 | $0.735 \pm 0.023$ | $0.797 \pm 0.008$ | $0.763 \pm 0.012$ | $0.808 \pm 0.007$ | $0.795 \pm 0.008$ | $0.788 \pm 0.010$ | $0.799 \pm 0.006$ | $0.769 \pm 0.017$ | $\mathbf{0.815} \pm 0.008$ | $\underline{0.813} \pm 0.012$ |
| H4ac | $0.275 \pm 0.022$ | $0.465 \pm 0.013$ | $0.564 \pm 0.011$ | $0.492 \pm 0.014$ | $0.502 \pm 0.025$ | $0.548 \pm 0.027$ | $0.585 \pm 0.018$ | $0.530 \pm 0.017$ | $\underline{0.592} \pm 0.015$ | $\mathbf{0.687} \pm 0.027$ |
| **Average MCC ($\uparrow$)** | 0.393 | 0.551 | 0.610 | 0.569 | 0.571 | 0.579 | 0.606 | 0.571 | $\underline{0.625}$ | **0.701** |
| *Regulatory Annotation* | | | | | | | | | | |
| Enhancer | $0.454 \pm 0.029$ | $0.525 \pm 0.026$ | $0.520 \pm 0.031$ | $0.545 \pm 0.028$ | $\underline{0.561} \pm 0.029$ | $0.522 \pm 0.024$ | $0.511 \pm 0.026$ | $0.516 \pm 0.018$ | $\mathbf{0.580} \pm 0.015$ | $0.558 \pm 0.011$ |
| Enhancer Type | $0.312 \pm 0.043$ | $0.423 \pm 0.018$ | $0.403 \pm 0.056$ | $0.444 \pm 0.022$ | $0.444 \pm 0.036$ | $0.403 \pm 0.028$ | $0.410 \pm 0.026$ | $0.433 \pm 0.029$ | $\underline{0.477} \pm 0.017$ | $\mathbf{0.519} \pm 0.005$ |
| Promoter All | $0.910 \pm 0.004$ | $0.945 \pm 0.003$ | $0.919 \pm 0.003$ | $0.951 \pm 0.004$ | $0.952 \pm 0.002$ | $0.937 \pm 0.002$ | $0.941 \pm 0.003$ | $0.926 \pm 0.004$ | $\underline{0.962} \pm 0.002$ | $\mathbf{0.967} \pm 0.013$ |
| Promoter NonTATA | $0.910 \pm 0.006$ | $0.944 \pm 0.003$ | $0.919 \pm 0.004$ | $0.969 \pm 0.003$ | $0.952 \pm 0.003$ | $0.935 \pm 0.007$ | $0.940 \pm 0.002$ | $0.925 \pm 0.006$ | $\underline{0.962} \pm 0.001$ | $\mathbf{0.971} \pm 0.007$ |
| Promoter TATA | $0.920 \pm 0.012$ | $0.911 \pm 0.011$ | $0.881 \pm 0.020$ | $0.919 \pm 0.008$ | $0.933 \pm 0.009$ | $0.895 \pm 0.010$ | $0.903 \pm 0.010$ | $0.891 \pm 0.009$ | $\underline{0.948} \pm 0.008$ | $\mathbf{0.961} \pm 0.015$ |
| **Average MCC ($\uparrow$)** | 0.701 | 0.750 | 0.728 | 0.766 | 0.768 | 0.738 | 0.741 | 0.738 | $\underline{0.786}$ | **0.796** |
| *Splice Site Annotation* | | | | | | | | | | |
| Splice Acceptor | $0.772 \pm 0.007$ | $0.909 \pm 0.004$ | $0.935 \pm 0.005$ | $\underline{0.973} \pm 0.002$ | $\underline{0.973} \pm 0.004$ | $0.918 \pm 0.017$ | $0.907 \pm 0.015$ | $0.912 \pm 0.010$ | $\mathbf{0.981} \pm 0.002$ | $0.969 \pm 0.013$ |
| Splice Site All | $0.831 \pm 0.012$ | $0.950 \pm 0.003$ | $0.917 \pm 0.006$ | $0.974 \pm 0.004$ | $\underline{0.975} \pm 0.002$ | $0.935 \pm 0.011$ | $0.953 \pm 0.005$ | $0.919 \pm 0.005$ | $\mathbf{0.976} \pm 0.011$ | $0.968 \pm 0.030$ |
| Splice Donor | $0.813 \pm 0.015$ | $0.927 \pm 0.003$ | $0.894 \pm 0.013$ | $0.974 \pm 0.002$ | $\underline{0.977} \pm 0.007$ | $0.912 \pm 0.009$ | $0.930 \pm 0.010$ | $0.888 \pm 0.012$ | $\mathbf{0.978} \pm 0.001$ | $0.960 \pm 0.007$ |
| **Average MCC ($\uparrow$)** | 0.805 | 0.929 | 0.915 | 0.974 | $\underline{0.975}$ | 0.922 | 0.930 | 0.906 | **0.979** | 0.965 |
| **Total Average MCC ($\uparrow$)** | 0.547 | 0.669 | 0.694 | 0.690 | 0.693 | 0.680 | 0.697 | 0.673 | $\underline{0.728}$ | **0.772** |
| **Total Average Rank ($\downarrow$)** | 9.67 | 6.72 | 6.00 | 4.83 | 4.56 | 6.33 | 5.61 | 7.22 | $\underline{2.06}$ | **1.67** |

GENERATOR by +0.044 MCC despite using only 14% of its parameters. The benchmark aggregates 18 datasets across three task families: (i) *histone mark prediction* from chromatin profiling, (ii) *regulatory annotation* (promoter and enhancer classification), and (iii) *splice-site annotation* at donor/acceptor boundaries. Gains are most pronounced on histone mark prediction, where DNACHUNKER improves average MCC by +0.076 (0.701 vs. 0.625) over GENERATOR, with per-dataset gains over the second-best baseline reaching +0.144 on H3K4me2 and +0.110 on H3K4me3. DNACHUNKER also leads on regulatory annotation (+0.010 average MCC) while remaining within 0.014 of the top splice-site result, indicating that the gains are broad rather than concentrated in a single task family. Notably, DNACHUNKER attains these results while trained solely on the human reference genome. Following Wu et al. (2025), we perform 10-fold cross-validation and report MCC per dataset and average rank across 10 models; baseline scores are taken from the same work, and finetuning details are deferred to Section C.1.

**Revised Nucleotide Transformer benchmark.** On the revised NT benchmark, DNACHUNKER achieves state-of-the-art performance, surpassing both DNA-targeted tokenization baselines MXDNA (Qiao et al., 2024) and PATCHDNA (Del Vecchio et al., 2026), shown in Table 2 with full per-task results in Table 10. Gains are most pronounced on splice-site annotation (+0.068 over MXDNA, +0.196 over PATCHDNA), a task family where prior work (Qiao et al., 2024; Lindsey et al., 2025) has re-

*Table 2.* **Revised Nucleotide Transformer Benchmark .** Due to space constraints, we show a subset of results on the revised Nucleotide Transformer Benchmark, restricted to learnable DNA-tokenization baselines. Values denote MCC (mean $\pm$ standard deviation) over 3 random seeds. Best **bold**, second best underlined. Full results are shown in Table 10

| | MxDNA | PatchDNA | DNACHUNKER |
|---|---|---|---|
| Histone markers | $0.555 \pm 0.020$ | $\underline{0.560} \pm 0.021$ | $\mathbf{0.562} \pm 0.020$ |
| Enhancers | $0.500 \pm 0.012$ | $\underline{0.512} \pm 0.009$ | $\mathbf{0.514} \pm 0.012$ |
| Promoters | $0.773 \pm 0.020$ | $\underline{0.806} \pm 0.011$ | $\mathbf{0.808} \pm 0.005$ |
| Splice site | $\underline{0.868} \pm 0.020$ | $0.740 \pm 0.028$ | $\mathbf{0.936} \pm 0.007$ |
| **Avg. MCC ($\uparrow$)** | $\underline{0.637}$ | 0.626 | **0.660** |

ported that segmentation-based tokenizers can underperform fixed BPE or k-mer schemes, due to task reliance upon uniform, fine-grained resolution. DNACHUNKER closes this gap while retaining the advantages of adaptive segmentation on histone-marker and regulatory-annotation tasks, suggesting that it preserves base-level sensitivity when needed without sacrificing broader contextual modeling elsewhere. We report MCC averaged over 3 random seeds, while baseline scores are taken from Del Vecchio et al. (2026). Additional finetuning details are deferred to Section C.2.

**Genomic benchmark.** On the Genomic Benchmarks suite, DNACHUNKER achieves the second best average rank and performs on par with GENERATOR in top-1 accuracy while using $7\times$ fewer parameters, and being trained only on the human reference genome unlike prior works (Wu et al., 2025; Dalla-Torre et al., 2025). The suite comprises nine

*Table 3.* **Genomic Benchmarks.** Performance on the Genomic Benchmarks suite. Values report top-1 accuracy (mean ± standard error) over 10-fold cross-validation. Higher is better for accuracy; lower is better for average rank. Best results are **bold**; second-best results are underlined.

| | DNABERT-2 (117M) | HyenaDNA (55M) | NT-v2 (500M) | Caduceus-PH (8M) | Caduceus-PS (8M) | GROVER (87M) | GENERator (1.2B) | GENERator-All (1.2B) | DNACHUNKER (172M) |
|---|---|---|---|---|---|---|---|---|---|
| Coding vs. Intergenomic | $0.951 \pm 0.002$ | $0.902 \pm 0.004$ | $0.955 \pm 0.001$ | $0.933 \pm 0.001$ | $0.944 \pm 0.002$ | $0.919 \pm 0.002$ | $\mathbf{0.963} \pm 0.000$ | $\underline{0.959} \pm 0.001$ | $0.955 \pm 0.012$ |
| Drosophila Enhancers Stark | $0.774 \pm 0.011$ | $0.770 \pm 0.016$ | $0.797 \pm 0.009$ | $\mathbf{0.827} \pm 0.010$ | $0.816 \pm 0.015$ | $0.761 \pm 0.011$ | $\underline{0.821} \pm 0.005$ | $0.768 \pm 0.015$ | $0.779 \pm 0.021$ |
| Human Enhancers Cohn | $0.758 \pm 0.005$ | $0.725 \pm 0.009$ | $0.756 \pm 0.006$ | $0.747 \pm 0.003$ | $0.749 \pm 0.003$ | $0.738 \pm 0.003$ | $\mathbf{0.763} \pm 0.002$ | $0.754 \pm 0.006$ | $\underline{0.761} \pm 0.011$ |
| Human Enhancers Ensembl | $0.918 \pm 0.003$ | $0.901 \pm 0.003$ | $0.921 \pm 0.004$ | $\mathbf{0.924} \pm 0.002$ | $\underline{0.923} \pm 0.002$ | $0.911 \pm 0.004$ | $0.917 \pm 0.002$ | $0.912 \pm 0.002$ | $0.922 \pm 0.007$ |
| Human Ensembl Regulatory | $0.874 \pm 0.007$ | $0.932 \pm 0.001$ | $\mathbf{0.941} \pm 0.001$ | $\underline{0.938} \pm 0.004$ | $\mathbf{0.941} \pm 0.002$ | $0.897 \pm 0.001$ | $0.928 \pm 0.001$ | $0.926 \pm 0.001$ | $0.935 \pm 0.005$ |
| Human non-TATA Promoters | $0.957 \pm 0.008$ | $0.894 \pm 0.023$ | $0.932 \pm 0.006$ | $\underline{0.961} \pm 0.003$ | $\underline{0.961} \pm 0.002$ | $0.950 \pm 0.005$ | $0.958 \pm 0.001$ | $0.955 \pm 0.005$ | $\mathbf{0.962} \pm 0.001$ |
| Human OCR Ensembl | $0.806 \pm 0.003$ | $0.774 \pm 0.004$ | $0.813 \pm 0.001$ | $\underline{0.825} \pm 0.004$ | $\mathbf{0.826} \pm 0.003$ | $0.789 \pm 0.002$ | $0.823 \pm 0.002$ | $0.812 \pm 0.003$ | $0.810 \pm 0.007$ |
| Human vs. Worm | $0.977 \pm 0.001$ | $0.958 \pm 0.004$ | $0.976 \pm 0.001$ | $0.975 \pm 0.001$ | $0.976 \pm 0.001$ | $0.966 \pm 0.001$ | $\mathbf{0.980} \pm 0.000$ | $\underline{0.978} \pm 0.001$ | $0.969 \pm 0.001$ |
| Mouse Enhancers Ensembl | $0.865 \pm 0.014$ | $0.756 \pm 0.030$ | $0.855 \pm 0.018$ | $0.788 \pm 0.028$ | $0.826 \pm 0.021$ | $0.742 \pm 0.025$ | $\underline{0.871} \pm 0.015$ | $0.784 \pm 0.027$ | $\mathbf{0.874} \pm 0.020$ |
| **Average Acc (↑)** | 0.876 | 0.846 | 0.883 | 0.880 | $\underline{0.885}$ | 0.853 | **0.892** | 0.872 | $\underline{0.885}$ |
| **Average Rank (↓)** | 5.11 | 8.22 | 4.17 | 3.89 | 3.33 | 8.11 | **2.89** | 5.44 | $\underline{3.29}$ |

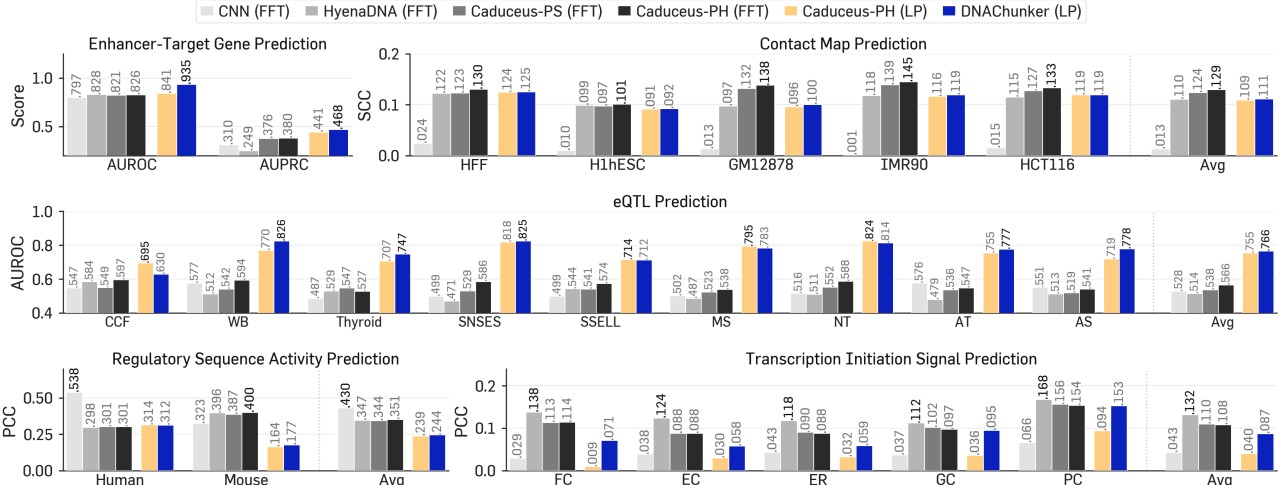

*Figure 2.* **DNALongBench.** Performance on DNALongBench across five long-range genomic prediction tasks. We compare DNACHUNKER and Caduceus-PH under linear probing (LP) with a frozen backbone, and include published full fine-tuning (FFT) baselines from Cheng et al. (2025) for reference. Values report the official metric for each subtask following the DNALongBench protocol. Higher is better. Best results are **bold**.

classification tasks, including enhancer and promoter recognition, coding vs. intergenic discrimination, and a human-vs-worm species control. Following Wu et al. (2025), we report top-1 accuracy averaged over 10-fold cross-validation; baseline scores are taken from Wu et al. (2025), and finetuning details are deferred to Section C.3.

**DNALongBench.** On DNALongBench, DNACHUNKER surpasses CADUCEUS (Schiff et al., 2024), the strongest DNA foundation model baseline reported in Cheng et al. (2025), on all five tasks (Figure 2). Gains are most pronounced on enhancer-target gene interaction (+0.061) and transcription initiation signal prediction (+0.047); on the former and on eQTL prediction, DNACHUNKER additionally surpasses the task-specific expert models reported in Cheng et al. (2025) despite using only linear probing. DNALong-Bench comprises five tasks probing long-range genomic dependencies with input contexts up to 1 Mb: (i) *enhancer-target gene interaction*, (ii) *expression quantitative trait loci (eQTL) prediction*, (iii) *contact map prediction*, (iv) *reg-*

*ulatory sequence activity*, and (v) *transcription initiation signal prediction*. Unlike Cheng et al. (2025), we freeze the backbone and train task-specific heads for 100 epochs with a learning rate of $1\mathrm{e}{-}3$. We additionally include the full-finetuned baseline scores for comparison, which were taken from Cheng et al. (2025). Specific details are deferred to Section C.4.

**BEND benchmark.** On the BEND benchmark, DNACHUNKER achieves the best average rank (1.9) across seven tasks, surpassing the previous best baseline PATCHDNA (2.1) (Table 4). DNACHUNKER ties for first on chromatin accessibility, histone modification, and CpG methylation, and leads on variant effect prediction for expression (0.59 AUROC), underscoring its ability to capture functional, noncoding regulatory signals. Unlike other benchmarks, BEND evaluates representations with a frozen backbone and a lightweight downstream head (Marin et al., 2024). BEND comprises seven downstream tasks on the human genome spanning three categories: long-range

*Table 4.* **BEND Benchmark.** Performance on the BEND benchmark across seven downstream tasks. Values report the official metric for each task following the BEND protocol. Higher is better for task metrics; lower is better for average rank. Best results are **bold**; second-best results are underlined.

| Metric | Gene finding | Enhancer annotation | Chromatin accessibility | Histone modification | CpG Methylation | Variant effects (expression) | Variant effects (disease) | Average Rank (↓) |
|---|---|---|---|---|---|---|---|---|
| | MCC | AUPRC | AUROC | AUROC | AUROC | AUROC | AUROC | – |
| NT-multi (2.5B) | **0.68** | **0.06** | 0.79 | 0.78 | **0.92** | 0.54 | 0.77 | 2.6 |
| NT-1000G (2.5B) | 0.49 | 0.04 | 0.77 | 0.77 | 0.89 | 0.45 | 0.49 | 6.9 |
| NT-v2 (500M) | 0.64 | 0.05 | 0.80 | 0.76 | 0.91 | 0.48 | 0.48 | 5.4 |
| DNABERT-2 (117M) | 0.43 | 0.03 | 0.81 | 0.78 | 0.90 | 0.49 | 0.51 | 5.6 |
| GENA-LM BERT (336M) | 0.52 | 0.03 | 0.76 | 0.78 | 0.91 | 0.49 | 0.55 | 5.1 |
| HyenaDNA (6.6M) | 0.35 | 0.03 | **0.84** | 0.76 | 0.91 | 0.51 | 0.45 | 5.7 |
| GROVER (87M) | 0.28 | 0.03 | 0.82 | 0.77 | 0.89 | 0.56 | 0.51 | 5.7 |
| PatchDNA (19.2M) | 0.58 | 0.04 | **0.84** | **0.79** | **0.92** | 0.51 | **0.84** | 2.1 |
| **DNACHUNKER** (172M) | 0.56 | 0.05 | **0.84** | **0.79** | **0.92** | **0.59** | 0.55 | **1.9** |

*Table 5.* **Ablation Study.** Linear probing performance on the revised NT benchmark. All models use the same architecture, training budget of 2B tokens, and either the GRCh38/hg38 or multispecies pretraining corpus. For BPE, Sanabria et al. (2024) is used. Values report Matthews correlation coefficient (MCC). Higher is better. Best results are **bold**.

| | Histone | Enhancers | Promoters | Splice | Overall |
|---|---|---|---|---|---|
| ***Human Reference Genome*** | | | | | |
| 6-mer | 0.338 | 0.319 | 0.593 | 0.147 | 0.347 |
| BPE | 0.339 | **0.349** | 0.667 | 0.223 | 0.375 |
| w/o Mask Protection | 0.316 | 0.293 | 0.614 | 0.128 | 0.332 |
| w/o Residual Gating | 0.338 | 0.298 | 0.607 | 0.185 | 0.353 |
| w/o Ratio Loss | 0.341 | 0.290 | 0.635 | 0.123 | 0.348 |
| **DNACHUNKER** | **0.344** | 0.346 | **0.673** | **0.290** | **0.390** |
| ***Multispecies*** | | | | | |
| BPE | 0.368 | 0.279 | 0.613 | 0.226 | 0.375 |
| **DNACHUNKER** | **0.443** | **0.362** | **0.730** | **0.242** | **0.448** |

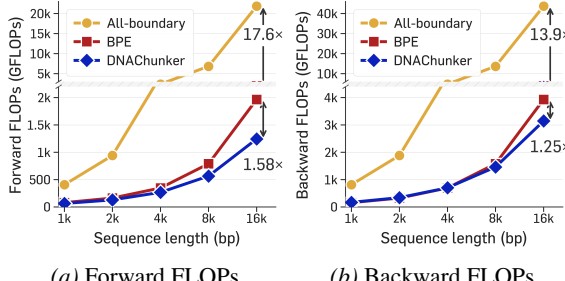

*(a)* Forward FLOPs      *(b)* Backward FLOPs

*Figure 3.* **Computation overhead.** Comparison of forward and backward computation overhead with models of same parameter size (170M), with different tokenization methods. For BPE, we use the tokenizer from Sanabria et al. (2024). Each curve reports the mean over 10 inputs per length (batch size 1).

annotation (gene finding, enhancer annotation), genome-scale epigenetic profiling (chromatin accessibility, histone modification, CpG methylation), and zero-shot noncoding variant effect prediction (expression and disease variants). We follow the BEND evaluation protocol and report per-task metrics alongside average rank; baseline scores are taken from Marin et al. (2024) and Del Vecchio et al. (2026), and evaluation details are deferred to Section C.5.

### 4.2. Ablative Studies

**Controlled ablation.** To isolate the performance benefits of each component, we perform a controlled ablation on a smaller setup by fixing dataset, training budget (2B tokens), and architecture (50M-parameter BiMamba → Transformer → BiMamba backbone). We evaluate the performance of each model on the Nucleotide Transformer revised benchmark via linear probing of 10 epochs with a learning rate of $1e-3$, where results are shown in Table 5. We find all three architectural components contribute: removing mask protection, residual gating, or the ratio loss each degrades average MCC, confirming them as necessary rather than incidental. Interestingly, we find that the advantage of adaptive tokenization further widens with data diversity: when trained on a multispecies corpus, DNACHUNKER improves by +0.058 over its HG38 counterpart and extends its lead

over BPE from +0.015 to +0.073, indicating that learned chunking benefits more from cross-species variation than fixed tokenization.

**Computation overhead.** Despite adding a learnable segmentation module on top of the transformer backbone, DNACHUNKER is the most compute-efficient among compared tokenization schemes in both forward and backward passes (Figure 3). At 16k-bp inputs, BPE incurs $1.58\times$ forward and $1.25\times$ backward FLOPs relative to DNACHUNKER, while single-base-pair tokenization is over an order of magnitude more expensive ($17.6\times$ forward, $13.9\times$ backward). The gap widens with sequence length, indicating that adaptive chunking compresses inputs more aggressively than fixed schemes as context grows — converting the overhead of learnable boundaries into a net efficiency gain at long contexts.

**Biological meaning of tokens.** To assess whether DNACHUNKER preserves functionally meaningful units, we measure how often biologically defined motifs are split across token boundaries. We sample 10,000 sequences of 8,192 bp from GRCh38/hg38 and locate occurrences of 29 transcription factor (TF) binding motifs from JASPAR 2024 (Rauluseviciute et al., 2024) alongside three cis-

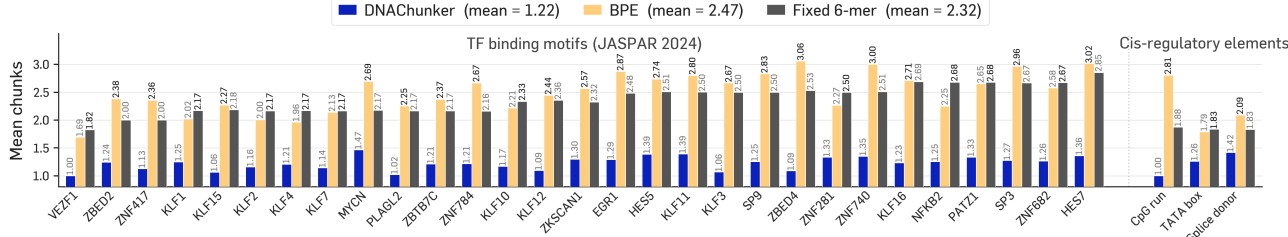

*Figure 4.* **Motif fragmentation analysis.** DNACHUNKER preserves motifs as single units. Mean chunks per occurrence for 29 JASPAR-2024 TF motifs and three cis-regulatory elements under our learned chunker, BPE (Sanabria et al., 2024), and fixed k-mer ($k = 6$) tokenization on matched 8 kb windows. Values near 1.0 indicate the motif is captured by a single chunk; larger values denote fragmentation.

*Table 6.* **Robustness of tokenizers against mutations.** Similarity scores between tokenizations of a reference sequence and its mutated counterpart on (i) ClinVar variants and (ii) GIAB HG002 variants. Higher is more stable under mutation. For BPE, we use the tokenizer from Sanabria et al. (2024). Best values are noted in **bold**.

| | ClinVar SNV | | ClinVar InDel | | GIAB HG002 Tier1 SV | | | | |
| | Benign | Pathogenic | Benign | Pathogenic | < 50 bp | 50–200 bp | 200 bp–1 kbp | 1–5 kbp | >5 kbp |
|---|---|---|---|---|---|---|---|---|---|
| BPE | **0.999** | **0.999** | 0.751 | 0.743 | 0.712 | 0.707 | 0.681 | 0.546 | 0.271 |
| **DNACHUNKER** (Stage 1) | **0.999** | **0.999** | **0.851** | **0.849** | **0.756** | **0.748** | **0.718** | **0.589** | **0.297** |
| **DNACHUNKER** (Stage 2) | 0.994 | 0.993 | 0.793 | 0.790 | 0.698 | 0.710 | 0.701 | 0.551 | 0.288 |

regulatory elements (CpG islands, TATA boxes, and splice donor sites), reporting the mean number of tokens each occurrence is fragmented into. Results are shown in Figure 4. DNACHUNKER fragments motifs into 1.22 tokens on average—keeping the majority intact as single tokens – whereas BPE averages 2.47 and is statistically indistinguishable from a sequence-blind fixed k-mer ($k = 6$) baseline (2.32), splitting nearly every motif across two to three tokens. The gap is consistent across both TF-binding and cis-regulatory categories, suggesting that BPE's frequency-driven merges do not align with functional boundaries, whereas DNACHUNKER learns to treat these motifs as coherent functional units rather than incidental substring patterns.

This motif-scale behavior extends to larger genomic regions (Section D): DNACHUNKER produces a clear chunk-length ordering by biological information density – short chunks over coding Exons ($\approx 14$–$16$ bp), longer ones over Promoters and Introns ($\approx 20$–$28$ bp), and the longest over SINEs, where length further decays with evolutionary age ($\approx 32 > 28 > 17$ bp for young / mid / old copies). The fixed BPE vocabulary cap cannot express this and collapses six of the seven categories into a single $\approx 10$ bp band.

**Robustness to mutations.** To test whether tokenizers produce *stable segmentations* under genetic perturbations, we compare the tokenizations of reference and mutated sequences across a spectrum of variant types and sizes. We sample 1,000 examples per category from ClinVar (Landrum et al., 2016) (benign and pathogenic SNVs and In-Dels) and additionally evaluate on structural variants from GIAB HG002 Tier1 SV v0.6 (Zook et al., 2020), stratified into five length bins from <50 bp to >5 kbp. For each

reference–variant pair, we compute

$$S(x^{\text{ref}}, x^{\text{mut}}) = (1 - \gamma) S_{\text{boundary}} + \gamma S_{\text{content}},$$

where $S_{\text{boundary}}$ is the Jaccard similarity between the sets of token boundary positions induced by the tokenizer and $S_{\text{content}}$ is a normalized edit-similarity over the token sequences themselves. We set $\gamma = 0.5$; higher values indicate more stable tokenization under mutation. Table 6 shows that DNACHUNKER remains more stable than BPE across nearly all variant types and length scales considered, from single-nucleotide substitutions through kilobase-scale structural rearrangements. This breadth suggests that the robustness arises from DNACHUNKER's context-dependent segmentation itself rather than from any specific variant regime, in contrast to BPE's fixed merges which are exposed to cascading shifts whenever a perturbation alters local substring frequencies.

## 5. Conclusion

DNA lacks the discrete "words" that anchor tokenization in natural language, making fixed schemes brittle. We frame tokenization as learnable and introduce DNACHUNKER, which jointly learns context-dependent adaptive segmentation and masked sequence representations. Across five benchmarks, DNACHUNKER improves over strong fixed-tokenization baselines while remaining parameter-efficient. Its learned segments show systematic structure: finer granularity in functionally enriched regions, coarser granularity in repetitive sequence, and stable boundaries under mutation. These results suggest that end-to-end, data-driven segmentation can yield genomic language models that better reflect biological organization.

## Acknowledgments

This work was supported by Institute for Information & Communications Technology Planning & Evaluation (IITP) grants funded by the Korea government (MSIT) (RS-2019-II190075, Artificial Intelligence Graduate School Program (KAIST); RS-2025-02304967, AI Star Fellowship (KAIST)), the GRDC (Global Research Development Center) Cooperative Hub Program through the National Research Foundation of Korea (NRF) grant funded by the Ministry of Science and ICT (MSIT) (No. RS-2024-00436165), the National Research Foundation of Korea (NRF) grants funded by the Korea government (MSIT) (RS-2025-02216257; No. RS-2022-NR072184; No. RS-2024-00406715; No. RS-2025-23523958), and Inocras Korea Inc. through a sponsored research collaboration.

## Impact Statement

DNACHUNKER's learnable adaptive segmentation offers a practical route to more faithful and efficient genomic representation learning: by allocating high resolution to functionally enriched regions (e.g., promoters/exons) while compressing repetitive sequence, it can improve performance on standard regulatory and splice-site benchmarks without requiring billion-parameter scale, potentially lowering the compute barrier for academic and clinical genomics research. At the same time, stronger DNA models can amplify risks around genetic privacy, re-identification, and downstream discriminatory use if applied to sensitive individual-level genomes; they may also contribute (indirectly) to dual-use biological design capabilities when combined with other tooling. We therefore recommend that deployments prioritize public/reference data or properly consented datasets, use access controls and auditing when handling human genomes, and restrict high-risk applications via institutional review and domain-specific oversight.

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

## A. Architecture Details

*Table 7.* Hyperparameters of DNACHUNKER architecture (171.59M parameters in total).

| Component | Architecture / Details | #Params |
|---|---|---|
| Token embedding | 10 vocab size 10; padded to 16, 640 dim | 10.24K |
| Encoder (Stage 1) | 2-layer BiMamba (bidirectional=True) | 5.61M |
| Router (Stage 1) | DifferentiableRoutingModule ($2\times$ Linear $640 \times 640$) | 819.20K |
| Encoder (Stage 2) | 2-layer BiMamba (bidirectional=True) | 5.61M |
| Router (Stage 2) | DifferentiableRoutingModule ($2\times$ Linear $640 \times 640$) | 819.20K |
| Main network | 30-layer TransformerBlock
• Attention: RoPE, 20 heads, 32 dim per head
• MLP: `mlp_mult` $= 4$ (hidden dim $= 2560$) | 147.50M |
| Decoder (Stage 1) | 2-layer BiMamba (bidirectional=True) | 5.61M |
| Decoder (Stage 2) | 2-layer BiMamba (bidirectional=True) | 5.61M |
| Dechunker 1 | BidirectionalGatedSmoothing | 640 |
| Dechunker 2 | BidirectionalGatedSmoothing | 640 |
| Final normalization | RMSNorm | 640 |
| Total | – | 171.59M |

Table 7 summarizes the architectural configuration of DNACHUNKER, comprising 171.59M parameters in total. The model follows a hierarchical encoder–decoder design with explicit routing modules and lightweight bidirectional smoothing components. The encoder is organized into two stages, each built from a 2-layer BiMamba backbone (5.61M params per stage) paired with a DifferentiableRoutingModule (819.20K params per stage), operating in a 640-dimensional representation space to produce routed query/key projections before passing the compressed representation downstream.

The main compute backbone is a 30-layer TransformerBlock (147.50M params) using RoPE attention with 20 heads and 32 dimensions per head. The main network accounts for most of the parameters. On the decoder side, DNACHUNKER applies two BiMamba stages (Stage 1/2; 5.61M each), followed by two extremely lightweight BidirectionalGatedSmoothing dechunkers and a final RMSNorm. Together, these components concentrate capacity in the Transformer trunk while keeping the hierarchical routing and smoothing pathways parameter-efficient.

## B. Pre-training Details

Table 8 summarizes the pretraining setup of DNACHUNKER, including dataset specifications, optimization strategy, masking details, and the repeat-downweighting protocol. We pretrain on the GRCh38/hg38 human genome using the curated 16k-window BED splits from the Enformer study, with sequences sampled at a fixed length of $2^{13}$ (8,192 bp). Each training step processes a token budget of $2^{20}$ tokens (per-device batch size of $32 \times 8,192$ tokens across 4 GPUs), and training proceeds for 50,000 optimizer steps, corresponding to roughly 52 billion processed base pairs.

Optimization is performed with the AdamW optimizer (Loshchilov & Hutter, 2019), using a peak learning rate of $1.25 \times 10^{-4}$, $\varepsilon = 1 \times 10^{-8}$, and weight decay of 0.01. We adopt a Warmup–Stable–Decay (WSD) schedule with 20% warmup, a stable plateau, and 20% linear decay; gradients are clipped to a global norm of 1.0. To balance the effective learning rate across the chunked stages of the model, we further apply a stage-wise learning-rate rule that scales each stage's LR by $\sqrt{B_s/B_{\mathrm{ref}}} \cdot \sqrt{D_{\mathrm{ref}}/D_s}$ to compensate for the differing batch and width of intermediate chunked representations. Pretraining is run in bfloat16 mixed precision.

For the differentiable chunking module, we set target retention ratios of 0.33 at stage 1 and 0.33 at stage 2, regularized by a compression-ratio auxiliary loss with weight 0.05 that pulls each stage toward its target budget.

Pretraining follows a masked language modeling objective with 15% of input nucleotides selected for corruption: 80% of these are replaced with a `[MASK]` token, 10% with a random base, and 10% left unchanged. Because raw human DNA contains a large fraction of repetitive elements (LINEs, SINEs, LTRs, simple/low-complexity repeats), naively weighting

*Table 8.* Pretraining details for the BiMamba + repeat-downweighted variant of DNACHUNKER.

| Category | Value |
|---|---|
| *Data* | |
| Reference genome | GRCh38 / hg38 |
| Interval source | Enformer splits |
| Sequence length | $2^{13} = 8{,}192$ bp (fixed) |
| Tokenizer | Per-nucleotide (vocab size 10) |
| MLM masking ratio | 15% |
| $\rightarrow$ [MASK] / random / unchanged | 80% / 10% / 10% |
| *Repeat down-weighting* | |
| Annotation | UCSC RepeatMasker `rmsk.hg38` |
| Non-repeat token weight | 1.0 |
| Repeat token weight | 0.1 |
| Applied to | Training loss only (val uses plain CE) |
| *Optimization* | |
| Optimizer | AdamW ($\varepsilon = 1 \times 10^{-8}$) |
| Peak learning rate | $1.25 \times 10^{-4}$ |
| Weight decay | 0.01 |
| LR schedule | WSD (warmup 0.2, decay 0.2) |
| Stage-wise LR rule | $\sqrt{B_s/B_{\text{ref}}} \cdot \sqrt{D_{\text{ref}}/D_s}$ |
| Gradient clip (global norm) | 1.0 |
| Precision | bfloat16 mixed |
| *Compute budget* | |
| Per-step token budget | $2^{20} = 1{,}048{,}576$ tokens |
| Per-device batch size | 32 sequences (8,192 bp each) |
| Devices | 4 GPUs |
| Optimizer steps | 50,000 |
| Total nucleotides processed | $\approx 5.2 \times 10^{10}$ bp |
| *Differentiable chunking* | |
| Target ratio (stage 1 / stage 2) | 0.33 / 0.33 |
| Compression-ratio loss weight | 0.05 |

all positions equally biases the loss toward easy-to-memorize repetitive motifs and away from informative regulatory and coding regions. To counter this, we use a per-token loss-reweighting scheme driven by the UCSC RepeatMasker annotation `rmsk.hg38` (Smit et al., 2013–2015): non-repeat nucleotides receive weight 1.0, while every RepeatMasker-annotated position is downweighted to 0.1. The weights are computed per sampled window via a pre-built repeat index, propagated through the collator alongside the padded inputs, and consumed by a weighted cross-entropy loss only at training time. This bidirectional, repeat-aware masking scheme encourages the model to leverage both local and global dependencies within DNA sequences while focusing capacity on biologically informative, non-repetitive regions.

## C. Finetuning Details on Downstream Tasks

### C.1. Nucleotide Transformer (NT)

We evaluate on the original Nucleotide Transformer downstream benchmark of Dalla-Torre et al. (2025), consisting of 18 sequence-classification tasks: three promoter variants (ALL/TATA/NO_TATA, 300 bp), two enhancer tasks (ENHANCERS, ENHANCERS_TYPES, 200 bp), three splice-site tasks (ALL/ACCEPTORS/DONORS, up to 600 bp), and ten histone-mark prediction tasks (H3, H4, H3K9AC, H3K14AC, H4AC, H3K4ME1/2/3, H3K36ME3, H3K79ME3, 500 bp). All inputs fit natively within the 8,192 bp pretraining context, so no sliding-window inference is required.

We adopt the *end-to-end fine-tuning* protocol of the original NT paper: the pretrained backbone is initialized from our MLM checkpoint and the entire network—backbone plus task head—is updated jointly. The task head is a masked-mean pooling layer over the valid non-padded token positions followed by a single linear classifier projecting to the task's class set. Sequences are tokenized with a single-nucleotide vocabulary $\{A, C, G, T, N\}$ augmented with [CLS], [SEP], [PAD], [MASK], and [UNK] special tokens. Following Wu et al. (2025), we evaluate using 10-fold cross-validation, and report Matthews correlation coefficient (MCC).

We use AdamW ($\beta_1 = 0.9$, $\beta_2 = 0.999$, $\varepsilon = 10^{-8}$) with weight decay 0.01, a constant learning-rate schedule without warm-up, gradient clipping at 1.0, and fp32 precision. For every task, we tune over a small grid of learning rates $\{1 \times 10^{-5}, 5 \times 10^{-5}, 1 \times 10^{-4}, 5 \times 10^{-4}\}$ and effective batch sizes $\{8, 16, 32, 64, 128\}$, yielding twenty $(\eta, B)$ configurations per task. Effective batch sizes are realized through gradient accumulation when the per-device micro-batch would otherwise exceed available memory. Training runs for up to 20 epochs with early stopping on the development metric using patience 5.

### C.2. Nucleotide Transformer Revised (NT Revised)

We evaluate on the NT Revised benchmark (Dalla-Torre et al., 2025), which relabels and curates the 18 NT tasks under a more biologically faithful protocol. The task categories mirror NT: three promoter, two enhancer, three splice-site, and ten histone-mark tasks, with histone substitutions H2AFZ, H3K27AC/ME3, H3K9ME3, and H4K20ME1 replacing several NT marks. Input lengths range from 300 bp for promoters to 1,000 bp for histone marks, all within the pretraining context.

We use the same end-to-end fine-tuning recipe as NT: pretrained backbone plus masked-mean pooling head and linear classifier, cross-entropy loss, MCC metric, single-nucleotide tokenization. Adhering to Del Vecchio et al. (2026) evaluation protocol, we use a 10% held-out development split (determined via random seed) from the official training partition.

We use AdamW ($\beta_1 = 0.9$, $\beta_2 = 0.999$, $\varepsilon = 10^{-8}$) with weight decay 0.01, a constant learning-rate schedule without warm-up, gradient clipping at 1.0, and fp32 precision. For every task, we tune over a small grid of learning rates $\{1 \times 10^{-5}, 5 \times 10^{-5}, 1 \times 10^{-4}, 5 \times 10^{-4}\}$ and effective batch sizes $\{8, 16, 32, 64, 128\}$, yielding twenty $(\eta, B)$ configurations per task. The configuration with the best development-set MCC is selected and its test-set MCC is reported. Effective batch sizes are realized through gradient accumulation when the per-device micro-batch would otherwise exceed available memory. Training runs for up to 20 epochs with early stopping on the development metric using patience 5.

### C.3. Genomics Benchmarks

We evaluate on the nine classification tasks of the Genomics Benchmarks suite (Grešová et al., 2023): DEMO_CODING_VS_INTERGENOMIC_SEQS (200 bp), DEMO_HUMAN_OR_WORM (200 bp), HUMAN_NONTATA_PROMOTERS (251 bp), HUMAN_ENHANCERS_COHN (500 bp), HUMAN_ENHANCERS_ENSEMBL (573 bp), HUMAN_OCR_ENSEMBL (593 bp), HUMAN_ENSEMBL_REGULATORY (802 bp, three classes), DROSOPHILA_ENHANCERS_STARK (3,237 bp), and DUMMY_MOUSE_ENHANCERS_ENSEMBL (4,776 bp). Each task is loaded from its dedicated HuggingFace dataset, and follow Wu et al. (2025) to evaluate with 10-fold cross-validation.

We again perform end-to-end fine-tuning with a masked-mean pooling head and a linear classifier, optimizing cross-entropy and reporting the accuracy.

We use AdamW ($\beta_1 = 0.9$, $\beta_2 = 0.95$, $\varepsilon = 10^{-8}$) with weight decay 0.1, gradient clipping at 1.0, and bf16-mixed precision. The learning rate is reduced on plateau using REDUCELRONPLATEAU with factor 0.95 and patience 1 epoch on the development metric. For every task, we tune over a small grid of learning rates $\{1 \times 10^{-5}, 5 \times 10^{-5}, 1 \times 10^{-4}, 5 \times 10^{-4}\}$ and effective batch sizes $\{8, 16, 32, 64, 128\}$, yielding twenty $(\eta, B)$ configurations per task. Effective batch sizes are realized through gradient accumulation when the per-device micro-batch would otherwise exceed available memory. Training runs for up to 20 epochs with early stopping on the development metric using patience 5.

### C.4. DNALongBench Benchmark

We evaluate on five DNALongBench (Cheng et al., 2025) tasks spanning enhancer–target gene prediction (ETGP; 450 kb, binary, AUROC), eQTL prediction across nine tissues (450 kb, binary, AUROC), contact-map prediction across five cell types (CMP; $\approx$1 Mb, regression, Pearson $r$), regulatory sequence activity prediction in human and mouse (RSAP; 196,608-bp TSS-centered input with predictions on the central 114,688 bp, tiled into 896 bins of 128 bp; Poisson regression, Pearson $r$), and transcription initiation signal prediction (TISP; 100 kb, regression, Pearson $r$). All five tasks are evaluated under the same *frozen-backbone linear-probe* protocol.

DNALongBench contains tasks whose inputs exceed the native 8,192 bp pretraining context. We adopt a *scatter–gather sliding-window* scheme:

- **Tiling.** The input of length $L$ is partitioned into overlapping windows of size $W = 8,192$ bp with stride $S = 4,096$ bp (50% overlap), yielding $\lceil (L - W)/S \rceil + 1$ windows. A final window anchored at $L - W$ is always appended so the right boundary is never dropped, regardless of divisibility.

- **Backbone forward pass.** Windows are flattened to a $(B \cdot N_{\text{win}}, W)$ tensor and run through the frozen backbone in sub-batches of at most 32 windows to bound activation memory; outputs are concatenated back to $(B, N_{\text{win}}, W, D)$.

- **Reaggregation.** For per-nucleotide tasks (TISP, CMP, RSAP), window-level hidden states are scattered back to their absolute genomic coordinates and divided by per-position coverage counts, yielding a clean $(B, L, D)$ embedding in which overlapping regions are arithmetically averaged. For pooled tasks (ETGP, eQTL), each window is collapsed to a single $D$-dimensional vector by mean pooling over valid tokens, producing $(B, N_{\text{win}}, D)$.

For RSAP, we first encode the full 196,608-bp TSS-centered input sequence. We then crop the central 114,688-bp prediction region, corresponding to 896 bins of 128 bp, mean-pool embeddings within each bin, and apply a linear projection to predict the assay tracks. The remaining 40,960 bp on each side serves as flanking context and is provided to the model as input but is not directly supervised. For eQTL, reference and alternate alleles are embedded under identical windowing, and the SNP feature is formed by concatenating $[\mathbf{h}_{\text{ref}}, \mathbf{h}_{\text{alt}}, \mathbf{h}_{\text{ref}} - \mathbf{h}_{\text{alt}}]$.

All probes are trained with AdamW, learning rate $1 \times 10^{-3}$, weight decay 0.1, 100 linear warm-up steps followed by cosine decay, gradient clipping 1.0, and bf16-mixed precision. The default batch size is 64, trained for up to 100 epochs with early stopping on the primary metric (patience 5). Losses are binary cross-entropy with logits for ETGP and eQTL, mean-squared error for CMP and TISP, and Poisson NLL for RSAP.

### C.5. BEND Benchmark

| Task | Length | Classes | Metric | Loss | Reference | Regime |
|------|--------|---------|--------|------|-----------|--------|
| Gene finding | 14,000 bp | 9 | MCC | CE | GRCh38 | supervised |
| Enhancer annotation | 100,096 bp | 1 | AUPRC | BCE (pw 82.86) | GRCh38 | supervised |
| Chromatin accessibility | 512 bp | 125 | AUROC | BCE | GRCh37.no-chr | supervised |
| Histone modification | 512 bp | 18 | AUROC | BCE | GRCh38 | supervised |
| CpG methylation | 512 bp | 7 | AUROC | BCE | GRCh38 | supervised |
| Variant effects (expression) | 512 bp | — | AUROC | — | GRCh38 | zero-shot |
| Variant effects (disease) | 512 bp | — | AUROC | — | GRCh38 | zero-shot |

*Table 9.* **BEND task specifications.** Supervised tasks train a two-layer CNN probe over frozen embeddings; variant-effect tasks score variants by the cosine distance between reference- and alternate-allele embeddings at the variant position.

We evaluate on the BEND benchmark (Marin et al., 2024), which includes five supervised representation-learning tasks and two zero-shot variant-effect tasks. We follow the official protocol of *frozen-embedding linear probing*: the pretrained backbone is never updated, and supervised learning is restricted to a lightweight task-specific head trained on cached embeddings. Details are provided in Table 9.

Our probe follows the official BEND configurations: a lightweight two-layer CNN consisting of two 1-D convolutions with kernel size 3, stride 1, padding 1, and GELU activations, followed by an optional task-specific average-pool window and a linear classifier. The benchmark uses hidden width 64 for gene finding, chromatin accessibility, histone modification, and CpG methylation. Outputs are downsampled by 128 bp for enhancer annotation and by 512 bp for chromatin accessibility, histone modification, and CpG methylation; gene finding is run at single-nucleotide resolution. Losses follow the benchmark: cross-entropy with ignore index $-100$ for padded positions in gene finding, and binary cross-entropy for the remaining tasks, including a positive-class weight of 82.86 for enhancer annotation.

All probes are trained according to the official BEND task configurations, using AdamW, weight decay 0.01, and a constant learning rate with no warm-up or scheduler. The learning rate is $3 \times 10^{-3}$ for gene finding, histone modification, chromatin accessibility, and CpG methylation, and $1 \times 10^{-3}$ for enhancer annotation. We train for up to 100 epochs with gradient clipping 1.0 and fp32 precision. Per-task batch sizes are 64 for gene finding, 256 for chromatin accessibility, histone modification, and CpG methylation, and 8 for enhancer annotation. Multi-label tasks are scored using per-class AUROC averaged across classes, the imbalanced enhancer annotation task is scored using AUPRC, and gene finding is scored using the multi-class Matthews correlation coefficient on unpadded positions.

## D. Additional Results

*Table 10.* **Revised Nucleotide Transformer Benchmark.** The reported values represent the Matthews Correlation Coefficient (MCC; mean $\pm$ standard deviation) averaged over 3 random seeds. Best results are **bold**; second best are underlined.

| | HyenaDNA (6.6M) | Caduceus-PS (7.7M) | GENA-LM-Base (110M) | NT-multi-100M (100M) | MxDNA (100M) | PatchDNA (19.2M) | DNACHUNKER (172M) |
|---|---|---|---|---|---|---|---|
| ***Histone Markers*** | | | | | | | |
| H2AFZ | $0.481 \pm 0.005$ | $0.507 \pm 0.007$ | $0.466 \pm 0.035$ | $0.501 \pm 0.009$ | $0.512 \pm 0.003$ | $\mathbf{0.523} \pm 0.010$ | 0.515 $\pm 0.002$ |
| H3K27ac | $0.440 \pm 0.003$ | $0.475 \pm 0.021$ | $0.495 \pm 0.010$ | 0.496 $\pm 0.009$ | $0.489 \pm 0.031$ | $0.486 \pm 0.015$ | $\mathbf{0.504} \pm 0.001$ |
| H3K27me3 | $0.554 \pm 0.014$ | $0.591 \pm 0.009$ | $0.588 \pm 0.004$ | 0.599 $\pm 0.009$ | 0.599 $\pm 0.015$ | $\mathbf{0.607} \pm 0.008$ | $0.590 \pm 0.007$ |
| H3K36me3 | $0.549 \pm 0.002$ | $0.607 \pm 0.008$ | $0.602 \pm 0.021$ | $0.617 \pm 0.004$ | $0.618 \pm 0.002$ | 0.621 $\pm 0.007$ | $\mathbf{0.623} \pm 0.005$ |
| H3K4me1 | $0.438 \pm 0.007$ | $0.471 \pm 0.014$ | $0.465 \pm 0.014$ | 0.487 $\pm 0.010$ | $\mathbf{0.497} \pm 0.001$ | $0.480 \pm 0.003$ | 0.487 $\pm 0.011$ |
| H3K4me2 | $0.523 \pm 0.025$ | $0.565 \pm 0.008$ | $0.538 \pm 0.027$ | $0.551 \pm 0.005$ | $0.563 \pm 0.012$ | $\mathbf{0.573} \pm 0.004$ | 0.572 $\pm 0.003$ |
| H3K4me3 | $0.618 \pm 0.007$ | $0.617 \pm 0.009$ | $0.610 \pm 0.055$ | $0.624 \pm 0.003$ | $0.627 \pm 0.017$ | 0.634 $\pm 0.005$ | $\mathbf{0.639} \pm 0.005$ |
| H3K9ac | $0.497 \pm 0.014$ | $0.526 \pm 0.009$ | $0.525 \pm 0.007$ | $0.531 \pm 0.002$ | $0.534 \pm 0.015$ | $\mathbf{0.569} \pm 0.010$ | 0.565 $\pm 0.007$ |
| H3K9me3 | $0.371 \pm 0.026$ | $0.435 \pm 0.015$ | $0.440 \pm 0.009$ | $0.469 \pm 0.006$ | $0.467 \pm 0.023$ | 0.470 $\pm 0.017$ | $\mathbf{0.485} \pm 0.009$ |
| H4K20me1 | $0.617 \pm 0.008$ | $0.639 \pm 0.009$ | 0.644 $\pm 0.011$ | $\mathbf{0.646} \pm 0.010$ | $\mathbf{0.646} \pm 0.007$ | $0.641 \pm 0.007$ | $0.640 \pm 0.011$ |
| ***Regulatory Annotation*** | | | | | | | |
| Enhancer | $0.479 \pm 0.005$ | $0.510 \pm 0.017$ | $0.483 \pm 0.023$ | $0.513 \pm 0.001$ | $0.519 \pm 0.014$ | 0.528 $\pm 0.009$ | $\mathbf{0.539} \pm 0.012$ |
| Enhancer type | $0.450 \pm 0.003$ | $0.471 \pm 0.006$ | $0.467 \pm 0.012$ | $0.478 \pm 0.002$ | $0.480 \pm 0.010$ | $\mathbf{0.496} \pm 0.008$ | 0.488 $\pm 0.011$ |
| Promoter all | $0.693 \pm 0.007$ | $0.742 \pm 0.010$ | $0.738 \pm 0.007$ | $0.737 \pm 0.019$ | $0.734 \pm 0.013$ | $\mathbf{0.791} \pm 0.009$ | 0.748 $\pm 0.007$ |
| Promoter non-TATA | $0.724 \pm 0.004$ | 0.764 $\pm 0.013$ | $0.736 \pm 0.025$ | $0.756 \pm 0.003$ | $0.755 \pm 0.010$ | $\mathbf{0.788} \pm 0.005$ | $0.760 \pm 0.003$ |
| Promoter TATA | $0.831 \pm 0.057$ | $0.761 \pm 0.028$ | $0.689 \pm 0.038$ | $0.818 \pm 0.052$ | $0.831 \pm 0.038$ | 0.840 $\pm 0.019$ | $\mathbf{0.916} \pm 0.005$ |
| ***Splice Site Annotation*** | | | | | | | |
| Splice acceptor | $0.820 \pm 0.015$ | $0.765 \pm 0.006$ | $0.760 \pm 0.005$ | $\mathbf{0.952} \pm 0.002$ | $0.812 \pm 0.032$ | $0.754 \pm 0.040$ | 0.947 $\pm 0.012$ |
| Splice site all | $0.849 \pm 0.006$ | $0.796 \pm 0.021$ | $0.764 \pm 0.013$ | $\mathbf{0.966} \pm 0.000$ | $0.860 \pm 0.007$ | $0.760 \pm 0.019$ | 0.937 $\pm 0.011$ |
| Splice donor | $0.840 \pm 0.029$ | $0.771 \pm 0.013$ | $0.781 \pm 0.004$ | $\mathbf{0.962} \pm 0.003$ | 0.931 $\pm 0.021$ | $0.706 \pm 0.026$ | $0.923 \pm 0.003$ |
| **Avg. MCC ($\uparrow$)** | 0.599 | 0.612 | 0.600 | 0.650 | 0.637 | 0.626 | **0.660** |
| **Avg. Rank ($\downarrow$)** | 6.06 | 4.89 | 5.67 | 3.06 | 3.06 | 3.00 | **2.05** |

**Revised NT Benchmark.** Table 10 reports the full results upon the revised NT Benchmark. We take the scores from the Del Vecchio et al. (2026). Note that DNACHUNKER achieves best average performance and best rank.

**Chunk size ablation.** To test whether DNACHUNKER's learned segmentation aligns with biological structure, we measure how its chunk length varies across functional and non-functional regions of the genome, and compare it against a fixed BPE tokenizer (GROVER (Sanabria et al., 2024), vocabulary 610, maximum token length 16 bp). We sample 2000 non-overlapping 8 kb windows per chromosome across chr1–22 and chrX, yielding $\approx 46$k windows and on the order of $6.5 \times 10^5$ annotated regions drawn from RefSeq (Goldfarb et al., 2025) (Exon, Promoter $= 2$ kb upstream of TSS, Intron) and RepeatMasker (Simple_repeat, and SINEs stratified by per-copy substitution divergence %div into young $<5$, mid $5-15$, and old $\geq15$). For each region $A$ we record $m_A = \max_{c\in\text{chunks}(A)} \text{len}(c)$, the length of the longest chunk whose center lies inside $A$, and report in Figure 5 the per-chromosome median of $m_A$ (annotations restricted to $|A| \geq 100$ bp; bars require $n \geq 30$). The lighter shading marks the $p_{95}$ of the same quantity.

DNACHUNKER produces a clear gradient ordered by biological information density: coding Exons and tandem Simple_repeats receive the shortest chunks ($\approx 14$–$16$ bp), Promoters and Introns sit one tier up ($\approx 20$–$28$ bp), and SINEs occupy the top of the figure, with the within-class divergence stratification recovering a monotone young $>$ mid $>$ old decay ($\approx 32 > 28 > 17$ bp) on every chromosome. The BPE baseline cannot resolve this gradient with comparable contrast: its outputs are confined between 6 and its 16 bp vocabulary cap, so young SINEs saturate the ceiling while the remaining six categories collapse into a $\approx 10$ bp band with no large between-class separation. DNACHUNKER, having no hard cap on token length, instead allocates extra length specifically to repeat-rich regions of low coding significance – the SINE/Simple_repeat side of the panel – while restricting itself to short chunks over Exons and Promoters where every base carries information. The same picture holds for the $p_{95}$ tail. The pattern is consistent across most chromosomes, indicating that the effect is driven by sequence-level statistics rather than sampling-specific artifacts.

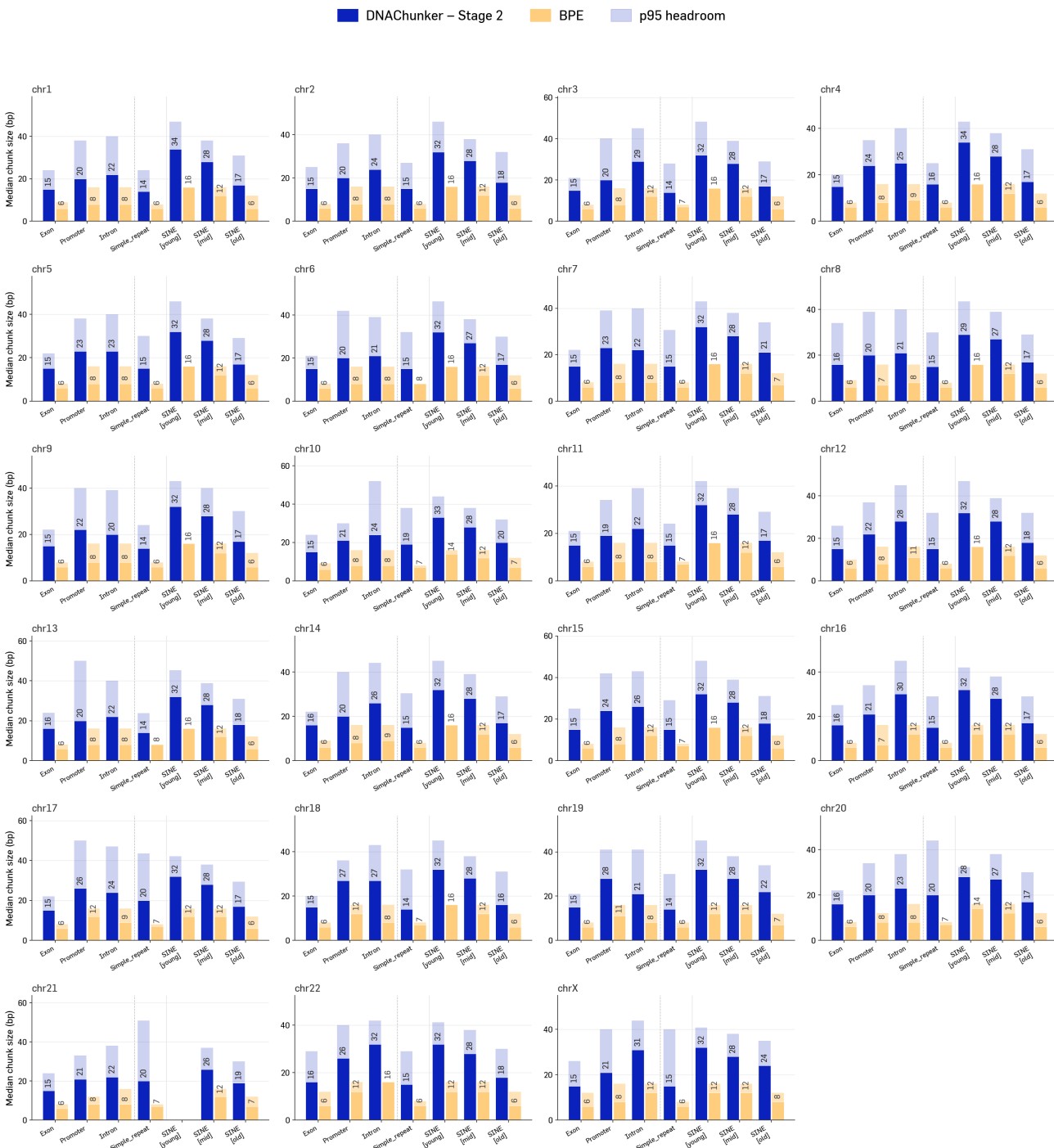

*Figure 5.* **Token size distribution per-chromosome.** Median chunk size by biological category, resolved per chromosome. Each panel shows one autosome (chr1–22) or chrX. Within a panel, seven categories are arranged by expected information density: protein-coding Exon, Promoter, Intron, Repeat, and three SINE cohorts stratified by RepeatMasker percent-divergence (young: %div < 5, mid: $5 \le$ %div < 15, old: %div $\ge$ 15). Solid bars report the median maximum chunk size (bp) produced by DNACHUNKER – Stage 2 (blue) and a BPE baseline (yellow); the lighter shading above each bar marks the $p_{95}$ ceiling and indicates the upper tail within that category. Dashed vertical guides separate functional from repeat categories, and young/mid/old SINEs.

