# OpenReview forum: "DNACHUNKER: Learnable Tokenization for DNA Language Models"
_ICML.cc/2026/Conference — ICML 2026 regular_

### Official Review · Reviewer_oYeT · 2026-03-09

**Soundness:** 2
**Presentation:** 3
**Significance:** 3
**Originality:** 1
**Overall Recommendation:** 3
**Confidence:** 4

**Summary:**

This paper introduces DNACHUNKER, a masked DNA language model that incorporates a learnable adaptive tokenization mechanism for genomic sequences. Instead of relying on fixed tokenization schemes such as single nucleotides, k-mers, or BPE, the model dynamically segments DNA into variable-length chunks conditioned on sequence context. The architecture uses a hierarchical encoder to perform adaptive segmentation, a Transformer-based main network operating on compressed segments, and a decoder that reconstructs base-pair–level representations.

The authors pretrain the model on the human reference genome (HG38) and evaluate it on two commonly used benchmarks: the Nucleotide Transformer benchmark and the Genomic Benchmark suite. The results suggest that DNACHUNKER achieves competitive or superior performance compared with several existing DNA language models, while using fewer parameters than some large baselines. The paper also provides analyses indicating that the learned segmentation tends to produce shorter chunks in biologically meaningful regions such as promoters and exons, and longer chunks in repetitive regions.

Overall, the work discusses the challenge of defining suitable tokenization strategies for genomic language models and proposes a learnable segmentation approach to address this issue. This work wants to examine the concept that tokenization itself should be learned jointly with sequence modeling rather than fixed a priori.

**Compliance With Llm Reviewing Policy:**

Affirmed.

**Key Questions For Authors:**

(1) To what extent do the improvements come from the adaptive tokenization itself versus other architectural choices (e.g., the hierarchical encoder-decoder design)? An ablation that keeps the architecture fixed but replaces the tokenizer with BPE or k-mers would help clarify this.

(2) Have the authors experimented with multi-species pretraining datasets or larger genomic corpora? It would be useful to know whether the adaptive tokenization behaves similarly when trained on more diverse sequences.

(3) The paper argues that adaptive chunking improves efficiency for long sequences. However, the experiments seem limited to sequences of length 8192 bp. How does the approach scale to substantially longer contexts (e.g., 100k+ bases)?

(4) The mutation robustness experiments mainly consider SNVs and small indels. Would the tokenization remain stable under larger structural variants or repeat expansions?

(5) The analysis suggests correlations between chunk length and genomic features. Are the learned tokens capturing recognizable biological motifs or regulatory patterns?

**Limitations:**

I don't see any potential negative societal impact in this paper.

**Strengths And Weaknesses:**

Strengths

(1) The paper addresses a real and underexplored issue in genomic language modeling: the lack of a natural tokenization unit in DNA sequences. Unlike natural language, DNA has no canonical word boundaries, and fixed tokenization schemes are not good choice under mutations or shifts. The paper makes a convincing case that tokenization deserves to be treated as a learnable component rather than a preprocessing step.

(2) The hierarchical encoder with adaptive boundary prediction is a technically interesting component. The use of cosine similarity–based boundary detection and two-stage compression is a relatively simple yet elegant mechanism for producing variable-length tokens. The mask-protection strategy for preventing tokenization artifacts in masked language modeling is also thoughtful and addresses a subtle issue in training.

(3) The model shows strong results on the Nucleotide Transformer benchmark, outperforming several baselines in terms of average MCC while using significantly fewer parameters than larger competing models.

Weaknesses

(1) While the paper claims to fill a gap by introducing learnable tokenization for masked DNA language models, the underlying idea is closely related to prior work on dynamic chunking. The main novelty seems to lie in adapting these ideas to the genomic MLM setting rather than introducing fundamentally new segmentation mechanisms.

(2) The experimental comparisons rely heavily on results reported from previous papers rather than re-running baselines under identical settings. This makes it difficult to determine whether improvements arise from the tokenization mechanism itself or from other training details (data processing, hyperparameters, or architecture differences).

(3) The model is pretrained only on the human reference genome. While this simplifies the experimental setup, it also limits the generality of the conclusions. Many recent genomic foundation models are trained on multi-species datasets or much larger corpora, which could influence both segmentation behavior and downstream performance.

(4) Although DNACHUNKER achieves strong results in certain tasks, improvements over the best baselines in the Genomic Benchmark suite appear relatively small in several cases. The gains are therefore not uniformly convincing across tasks.

(5) The paper includes some comparisons with other tokenization approaches, but more detailed ablations would strengthen the claims. For example:

 - How sensitive are results to the target compression ratio?

 - How much improvement comes from segmentation vs. the architectural changes around it?

 - What happens if the main network operates at the base-pair level with the same architecture?

---

> ### Author Rebuttal · Authors · 2026-03-31
>
> We thank the reviewer for their constructive feedback. We are glad the reviewer recognizes our exploration of the "real and underexplored issue" of genomic tokenization, with "elegant" design of the encoder module.
>
> We provide tables and figures here: https://anonymous.4open.science/r/dnachunker-rebuttal-5943/reviewer_oYeT/README.md, and address each concern in detail below.
>
> **W1: Limited novelty.** We respectfully disagree. Prior dynamic chunking methods are inherently unidirectional; this is a fundamental architectural assumption, not an implementation detail. DNA has no canonical reading direction; regulatory motifs act in both orientations, and functional prediction requires upstream and downstream context. Our bidirectional chunking mechanism is a structurally different segmentation procedure, not a re-packaging of existing ones.
>
> Beyond bidirectionality, we identify and solve two failure modes unique to combining adaptive segmentation with MLM, neither discussed in prior work:
> * Mask-conditioned segmentation. [MASK] tokens are pretraining artifacts absent at inference. Without isolation, the routing network learns boundaries based on mask placement — a spurious signal that does not transfer. Our mask protection ensures segmentation reflects genomic context only.
> * Residual information leakage. Bidirectional encoders mix neighbor information, so standard residuals let masked tokens be reconstructed from leaked encoder context, bypassing the main network. Our masked residual gating prevents this shortcut.
>
> These address structural incompatibilities between adaptive segmentation and bidirectional MLM. The ablation in Table 1 confirms both are essential: removing either drops performance below the fixed BPE baseline, highlighting the necessity of both designs.
>
> **W2, W5, Q1: Baselines not re-run under identical conditions, Insufficient ablations.** To address these concerns, we fix the model backbone and perform the following ablations.
> * Segmentation vs. architecture (Table 2). Under identical architecture and training conditions, DNAChunker achieves the best overall MCC (0.3902 vs. 0.3753 for BPE), with gains most pronounced on splice sites (+0.067 over BPE). This confirms that the improvements stem from the adaptive tokenization mechanism itself, not from confounding architectural or training details.
> * Component analysis (Table 1). In the same controlled setup, we ablate each component of DNAChunker individually. Without mask protection or residual gating, the model degrades below the BPE baseline, highlighting the necessity of both contributions.
> * Sensitivity to target compression ratio α (Table 3). We vary the target ratio α ∈ {0.33, 0.5, 0.8}. All three settings outperform the BPE baseline in overall MCC, indicating that the gains are not an artifact of a single tuned hyperparameter.
>
> **W4: Gains not uniformly convincing.** To strengthen the evidence, we extended our evaluation during the rebuttal period to two recent benchmarks: BEND (Table 4) and DNALongBench (Table 5~9). DNAChunker achieves the best average rank on BEND (2.0 vs. 2.3 of PatchDNA) and consistent improvements on DNALongBench over Caduceus (the most recent benchmark shown in DNALongBench). Across all 6 benchmarks, DNAChunker ranks top-1 or top-2 in every suite.
>
> Importantly, these gains are attributable to the learnable tokenization itself, as outlined in response to W2. We also note that no model in this field achieves uniform gains across all subtasks. Even GENErator (1.2B, previous SOTA) loses to the 55M HyenaDNA on 5 of 10 histone tasks in the NT Benchmark. What distinguishes DNAChunker is its consistency in aggregate across suites, achieved with 7× fewer parameters and training on the human genome alone.
>
> **Q2, W3: Human-genome-only pretraining.** We perform an ablative study to test the gains of utilizing multi-species data. In Table 10, we find DNAChunker amplifies the advantage rather than being hindered by the cross-species diversity. Specifically, the gap over BPE widens from +0.015 (vs. Human BPE) to +0.073 (vs. MS-BPE).
>
> **Q3: Long-context scaling**
> We evaluated DNAChunker on DNALongBench (Table 5~9) and find it outperforms Caduceus on all task families, confirming that learned tokenization transfers to long-range genomic tasks.
>
> **Q4: Structural variant robustness.** We evaluated tokenization stability utilizing the GIAB HG002 Tier1 SV benchmark, shown in Table 11. DNAChunker consistently outperforms BPE in maintaining tokenization integrity on large genetic variations up to 5k+ base pairs.
>
> **Q5: Biological motif alignment.** We provide empirical evidence that DNAChunker captures known motifs. We measured the mean number of chunks spanning each motif instance across 29 TF binding motifs and 3 cis-regulatory elements, shown in Figure 1. DNAChunker averages 1.22 chunks per motif, compared to 2.47 (BPE) and 2.32 (6-mer), indicating that it largely preserves motifs within a single chunk rather than fragmenting them.

---

### Official Review · Reviewer_Kj4M · 2026-03-12

**Soundness:** 2
**Presentation:** 3
**Significance:** 4
**Originality:** 4
**Overall Recommendation:** 3
**Confidence:** 4

**Summary:**

Overall, the work discusses the challenge of tokenizing genomic sequences, where canonical boundaries, unlike "words" in natural language, do not inherently exist. Fixed tokenization schemes (like k-mers or BPE) are brittle to mutations and fail to capture variable-length functional elements. To address this, the authors propose DNACHUNKER, a bidirectional masked language model (MLM) featuring a learnable, dynamic segmentation module. This work strives to examine the concept of context-dependent tokenization within a non-autoregressive paradigm by utilizing a hierarchical architecture (BiMamba encoder/decoder with a Transformer main network) and a specific "mask protection" mechanism. Pre-trained on the human reference genome, DNACHUNKER achieves highly competitive results on the Nucleotide Transformer and Genomic Benchmarks. Furthermore, qualitative analyses show the model learns biologically meaningful chunking strategies, allocating finer granularity to coding/conserved regions and coarser chunks to repetitive elements.

**Compliance With Llm Reviewing Policy:**

Affirmed.

**Key Questions For Authors:**

1) Architectural Ablation: Can you provide the downstream performance of your exact base architecture (BiMamba encoder → 30-layer Transformer → BiMamba decoder) trained with a fixed BPE tokenizer instead of the DifferentiableRoutingModule? This is critical to prove that your performance gains stem from the dynamic chunking rather than the hybrid backbone.

2) Data Contamination: How did you handle the overlap between your pre-training corpus (HG38 Enformer splits) and the test sets of the Genomic and Nucleotide Transformer benchmarks? Were the specific downstream test contigs strictly filtered out from the pre-training data?

3) Computational Overhead: What is the empirical wall-clock time and memory consumption for processing an 8192-bp sequence with DNACHUNKER compared to a standard BPE-based Transformer of equivalent parameter count?

4) Mask Protection: Is the "Mask protection mechanism" strictly necessary? Do you have empirical results showing the failure modes (e.g., boundary collapse) if the model is trained without this constraint?

5) Benchmark Discrepancy: Why are the DNA-targeted baselines (MxDNA, PatchDNA) evaluated on a "Revised" NT Benchmark in Table 3, while the main results in Table 1 use the older version of the benchmark?

**Limitations:**

The authors do not adequately discuss the limitations of their work in the main text. There is no mention of the computational and memory overhead induced by running bidirectional Mamba layers and cosine-similarity routers at the raw base-pair resolution before compression. Furthermore, the potential for data leakage (evaluating on human genome tasks after pre-training on the human genome) is a critical limitation that is entirely omitted from the discussion. A dedicated "Limitations" paragraph must be added to address these computational and data-provenance constraints.

**Strengths And Weaknesses:**

**Soundness:**

- *Strengths:*

The structural analysis of the learned tokenization (Figures 1c, 2, and 3) is a major strength of this paper. Demonstrating empirically that the differentiable router naturally learns to compress repetitive SINE regions while preserving base-pair-level resolution for exons, introns, and promoters is a compelling validation of the core hypothesis. Furthermore, the analysis of tokenizer robustness against SNVs and InDels (Table 4) provides a concrete, measurable advantage over static BPE.

- *Weaknesses:*

The paper suffers from two severe methodological omissions that undermine the claim that the dynamic tokenizer is responsible for the performance gains:

*Confounded Architecture and Missing Ablations:* DNACHUNKER introduces a highly customized, powerful hybrid architecture consisting of bidirectional Mamba layers, differentiable routing modules, and a 30-layer RoPE-Transformer trunk. However, there is no ablation study isolating the effect of the tokenization mechanism itself. To scientifically claim that learnable tokenization drives the state-of-the-art performance, the authors MUST evaluate a baseline using the exact same BiMamba+Transformer architecture but equipped with a standard, fixed BPE or k-mer tokenizer. Without this, the performance gains in Tables 1 and 2 could stem entirely from the expressive power of the 30-layer Transformer and Mamba feature extractors rather than the dynamic chunking.

*Unaddressed Pre-training Data Contamination:* The model is pre-trained on the HG38 human reference genome using the "Enformer study splits" (which typically hold out chromosomes 8, 9, and 10). However, the downstream evaluations (Nucleotide Transformer and Genomic Benchmarks) consist of human genomic sequences scattered across the entire genome. If the downstream test sets were not strictly filtered out of the HG38 pre-training corpus, DNACHUNKER has effectively seen the test data during unsupervised pre-training. Comparing its performance against baselines (e.g., GENErator or NT) that may have utilized different pre-training corpora or rigorous deduplication pipelines creates an unfair evaluation.

*Missing Mask-Protection Ablation:* In Section 3.1.1, the authors introduce a "Mask protection mechanism" to prevent the model from exploiting [MASK] tokens as boundary shortcuts. While theoretically sound, there is no empirical ablation demonstrating what happens if this protection is removed. Does the model actually collapse?

**Presentation:**

- *Strengths:*

The motivation is clearly articulated, and the architectural diagrams (Figure 1a) effectively communicate a complex multi-stage pipeline.

- *Weaknesses:*

The paper lacks a dedicated discussion on computational overhead. Computing cosine similarities at base-pair resolution across 8192-length sequences, followed by a 30-layer Transformer, incurs a significant cost. The authors must provide a FLOPs or wall-clock inference latency comparison against a standard BPE-based Transformer of similar size.

Minor typographical errors exist (e.g., referring to "BPEe" instead of "BPE" in Appendix B, Line 590).

In Table 3, the authors evaluate against MxDNA and PatchDNA on the "Revised" NT benchmark, but Table 1 uses the "Original" NT benchmark. The rationale for this split evaluation is unexplained and makes cross-referencing confusing.

**Significance:**

- *Strengths:*

Tokenization is currently one of the most critical bottlenecks in genomic foundation models. Successfully demonstrating an end-to-end learnable tokenizer for MLMs that correlates with biological taxonomy (exons vs. repeats) is a highly significant step forward for the field of computational biology.

**Originality:**

- *Strengths:*

While learnable tokenization exists in autoregressive text models (e.g., Byte Latent Transformer, H-Net), adapting this concept to a bidirectional masked modeling framework is highly non-trivial. The design of the hierarchical BiMamba encoder, combined with the bidirectional gated smoothing decoder and explicit mask-boundary protection, is exceptionally novel and well-engineered.

---

> ### Author Rebuttal · Authors · 2026-03-31
>
> We thank the reviewer for their thorough and constructive feedback. We are glad the reviewer considers our structural analysis "a major strength" providing "a compelling validation of the core hypothesis," and our robustness analysis "a concrete, measurable advantage over static BPE."
>
> We provide tables and figures in the following link:https://anonymous.4open.science/r/dnachunker-rebuttal-5943/reviewer_Kj4M/README.md.
>
> Below we address each concern in detail.
>
> **W1, Q1: Architectural ablation.** We agree this ablation is critical. Thus, we trained three variants differing only in tokenization: (1) fixed 6-mer, (2) BPE, and (3) DNAChunker, with the routing module removed for fixed-tokenizer baselines. As shown in Table 1, DNAChunker consistently outperforms both, with large gains on splice sites (+0.067 over BPE), confirming that improvements are driven by the chunking mechanism, not due to architectural difference.
>
> **W2, Q2: Data contamination risk.**
> We acknowledge that DNAChunker's pretraining data includes chromosomes 20 and 21 used by the NT benchmark test set.
> **However, we emphasize two points: (1) this overlap is a field-wide condition shared by virtually all baselines with DNAChunker being one of the most constrained and (2) DNAChunker dominates over all the models trained on the Enformer split.**
>
> We summarize the data contamination issues of the algorithms:
> - Enformer, HyenaDNA, Caduceus, PatchDNA, MxDNA: Same Enformer split as DNAChunker, with identical chromosome-level overlap with the NT benchmark. No multi-species data, so no overlap with multi-species tasks in the Genomic Benchmark (e.g., Drosophila, mouse, worm).
> - DNABERT-2, GENErator: Pretrained on multiple species including the complete human genome, with no documented exclusion of NT test chromosomes. Has potential overlap with multi-species tasks in the Genomic Benchmark (mouse, Drosophila).
> - NT: The only model with managed separation on its own benchmark, yet its multi-species pretraining corpus (NT-MS) likely overlaps with Genomic Benchmark species.
>
> Beyond this comparison, we emphasize that DNAChunker consistently outperforms all baselines using the same Enformer split across every benchmark dataset—both those reported in the main paper and those added during the rebuttal, including BEND (Table 6) and DNALongBench (Tables 7–11). This is further supported by ablation results under a controlled setup (Table 1), where DNAChunker surpasses fixed tokenization baselines.
>
> In summary, DNAChunker uses only a single species and a single reference assembly (HG38), making it among the most constrained pretraining setups in the comparison. It has strictly less potential contamination on the Genomic Benchmark than multi-species models such as GENErator, DNABERT-2, and NT-MS. Moreover, all models in this comparison, including DNAChunker, **use self-supervised objectives without access to downstream task labels** (masked or autoregressive language modeling) during pretraining. The overlap is therefore at the level of unlabeled sequence, not supervised signal.
>
> We agree this is a meaningful consideration for the field and will add a discussion of pretraining/benchmark overlap and chromosome-level held-out splits in the revised paper.
>
> **W3,  Q4: Mask protection necessity & ablation.** In Table 2, we empirically validate that mask protection and residual gating are crucial: removing them degrades all NT-revised subtasks, with overall MCC dropping by 0.058 and splice sites suffering most.
>
> **W4, Q3: Computational cost.** In Table 4 and 5, we report wall-clock time, peak GPU memory, and FLOPs compared with a BPE transformer of similar parameter size (details specified in Table 3). We find the gains of token compression is much larger than additional compute from the routing module, and DNAChunker achieving better efficiency overall.
>
> **W5, Q5: Table split rationale.** To clarify, the original NT benchmark and the revised NT benchmark are separate collections of datasets, not different splits of the same data. Thus, Tables 1 and 3 of main table evaluate on entirely non-overlapping tasks.
>
> We present the results of  the revised benchmark because the most recent tokenization baseline (PatchDNA) has not released checkpoints and has only evaluated on the revised NT benchmark. Given the direct topical relevance, a head-to-head comparison was necessary and thus included. Upon their checkpoint release, we will compare against them on the original NT benchmark as well. We note that DNAChunker achieves the best average MCC and rank on both benchmarks.
>
> **W6: Typo.**
> We thank the reviewer for pointing this out. We will revise it in the new manuscript.
>
> **W7: No limitations section.**
> We will add a limitations section in the revised manuscript, including (1) lack of full multi-species training despite its evidenced gains in ablations, (2) additional architectural complexity, and (3) fixation of ratio loss regardless of context.

---

### Official Review · Reviewer_QmWN · 2026-03-12

**Soundness:** 3
**Presentation:** 3
**Significance:** 2
**Originality:** 3
**Overall Recommendation:** 5
**Confidence:** 4

**Summary:**

The paper introduces DNAChunker, a masked DNA language model that uses dynamic chunking after a first encoder layer to more efficiently process long sequences in a transformer stack.

**Compliance With Llm Reviewing Policy:**

Affirmed.

**Final Justification:**

The rebuttal thoroughly addressed my main concerns.

**Key Questions For Authors:**

See weaknesses for critical questions; my main concern is more comprehensive benchmarking to establish long-range performance, as this is a claimed focus of the work.


Minor clarification questions:
- Is it necessary for the first-stage encoder layer to work globally, wouldn’t a local receptive field be enough?
- I do not fully understand why a mask must strictly form a singleton chunk. A masked nucleotide may well be part of a functional chunk, and as masks are placed at random I don’t follow the argument that the model would excessively condition its chunking on mask presence. Could you clarify?

**Limitations:**

The paper would benefit from discussing limitations in the conclusion. Right now, this is not the case.

**Strengths And Weaknesses:**

**Strengths**

1) The chunking mechanism is an original contribution that seems well-adapted to the requirements of the DNA domain.
2) The unsupervised chunking behaviour is demonstrated to be biologically sensible.

**Weaknesses**

1) Lack of good benchmarks. There has been a flurry of research showing that neither the Nucleotide Transformer benchmark nor Genomic benchmarks are particularly biologically relevant, and are especially unsuited to long context evaluations. See e.g. BEND, Marin et al 2023 or DNALongBench, Cheng et al 2025.
2) The model is only trained on the human genome. NT and Evo have shown that multi-genome training can be advantageous as it provides evolutionary supervision - this should be discussed I suppose the arising chunking behavior may also differ with species?
3) Evaluation under mean pooling only. This may make it especially easy for BPE/fixed k-mer models to suffer as the more numerous embeddings from unimportant parts may dominate the mean, whereas the adaptive chunking as I understand it may "concentrate" the relevant information in less embeddings. Is this a concern, and would the gains still hold if a downstream model (e.g. a CNN) operated on the actual sequence-length embedding matrix?

---

> ### Author Rebuttal · Authors · 2026-03-31
>
> We thank the reviewer for the constructive and thorough review. We appreciate the recognition of (1) the originality of our learnable chunking mechanism and its adaptation to the DNA domain, and (2) the biological sensibility of the unsupervised chunking behavior.
>
> We provide tables and figures in the following link: https://anonymous.4open.science/r/dnachunker-rebuttal-5943/reviewer_QmWN/README.md, and address each concern in detail below.
>
> **W1: Benchmarks are not biologically relevant, & Q3: Long-range benchmarking.** To address the concerns, we extend our evaluation to include *both* BEND (Table 1) and DNALongBench (Table 2–6). On BEND, DNAChunker achieves the best average rank (2.0) across all 7 task categories, outperforming PatchDNA (2.3) and NT-MS (2.6). On DNALongBench, DNAChunker surpasses Caduceus on all 5 subtasks, with particularly large gains on ETGP (+0.061) and TISP (+0.047, more than doubling performance). Notably, on ETGP and eQTLP, our linear probing results surpass even the expert models in the original DNALongBench paper. These results confirm that DNAChunker's advantages extend to biologically grounded, long-range evaluation suites.
>
> **W2: Lack of multi-genome training.** In the main paper, we focused on the human genome, aligning with the setup of prior works (Caduceus, Enformer, MxDNA). To test generalizability to multi-species, we vary only the pretraining dataset (HG38 human dataset vs. NT-MS multispecies dataset) under the shared controlled setup. Results are shown on Table 7. DNAChunker benefits from multispecies training substantially more than fixed BPE: a +0.058 overall gain (0.3902→0.4482) compared to BPE's flat +0.0 (0.3753→0.3753). This demonstrates that context-dependent tokenization better exploits the cross-species evolutionary variation.
>
> **W3: Mean pooling may favor adaptive chunking unfairly.** We first note that mean pooling followed by a linear layer is the standard evaluation protocol shared by Caduceus, Nucleotide Transformer, DNABERT-2, and HyenaDNA. Nevertheless, we directly test the reviewer's concern with a CNN-based probing ablation under the shared controlled setup in Table 8. CNN probing does improve BPE's performance (as the reviewer anticipated). However, such gain holds for DNAChunker as well, showing best overall MCC performance. This confirms that the gains from adaptive chunking are not an artifact of a specific pooling strategy.
>
> **W4: No limitations discussion.** We will add a limitations section in the revised manuscript, including (1) lack of full multi-species training despite its evidenced gains in ablations, (2) additional architectural complexity, and (3) fixation of ratio loss regardless of context.
>
> **Q1: Global vs local encoder.** There is no strict need for the encoder to have a global receptive field, and can be modeled with a local one, *i.e.* CNN. We  directly ablate this - comparing a CNN encoder (kernel size 7, local context) against BiMamba (full context) under the shared controlled setup, shown in Table 9. The CNN encoder outperforms BiMamba on Histone Marks, which is expected as histone modifications are primarily governed by local sequence context. In contrast, BiMamba achieves a notably larger gain on Splice Sites (+3.6pp over CNN). Overall, BiMamba leads (0.3902 vs 0.3885).
>
> **Q2: Mask singleton chunk justification.** The reviewer raises a fair point that a masked nucleotide could belong to a functional chunk. However, allowing masks to merge with neighbors lets the routing network exploit the [MASK] signature as a boundary cue absent at test time. Mask protection prevents this: forced boundaries at masked positions bypass the routing network entirely, so its parameters are never trained on [MASK]-adjacent signals and are shaped exclusively by genomic context. The forced boundaries disappear at test time along with the masks, which is safe because no learned parameters ever depended on them.
>
> We validate this empirically upon a controlled experiment scheme, shown in Table 10. Without our mask protection mechanism the performance drops significantly, demonstrating the necessity of the mask protection mechanism.

---

> > ### Author Rebuttal · Reviewer_QmWN · 2026-04-04
> >
> > Thank you for the thorough rebuttal!

---

> > > ### Author Response · Authors · 2026-04-04
> > >
> > > We are pleased that our rebuttal has satisfactorily addressed your concerns, and we appreciate your positive assessment of our work.

---

### Official Review · Reviewer_WZ97 · 2026-03-13

**Soundness:** 3
**Presentation:** 3
**Significance:** 3
**Originality:** 3
**Overall Recommendation:** 5
**Confidence:** 4

**Summary:**

This paper proposes DNAChunker, a masked DNA language model with a learnable adaptive tokenization / segmentation module that produces variable-length chunks instead of relying on fixed single-nucleotide, k-mer, or BPE tokenizations. The model uses a hierarchical encoder to infer chunk boundaries, a Transformer main network over the compressed sequence, and a hierarchical decoder to reconstruct base-pair representations. The authors report strong results on the Nucleotide Transformer and Genomic Benchmarks, and provide analyses suggesting that learned chunks are shorter in functionally enriched regions and longer in repetitive regions. Overall, the work discusses the challenge of choosing biologically appropriate tokenization for genomic language models, and this work strives to examine the concept of end-to-end learned segmentation in the masked DNA modeling setting.

**Compliance With Llm Reviewing Policy:**

Affirmed.

**Final Justification:**

Weaknesses addressed. Additional results are informative.

**Key Questions For Authors:**

1. Can you clarify why encoder-context “leakage” is problematic here but not simply equivalent to standard bidirectional MLM behavior?

2. Can you better justify the biological claim about frequency-driven tokenization failing to capture functional importance? The motivation is plausible, but terms such as “substrings” and “functional importance” are left somewhat vague.

**Limitations:**

yes

**Strengths And Weaknesses:**

Strengths

This work strives to examine the concept of adaptive, context-dependent tokenization in genomic language modeling, and that is a worthwhile direction. The paper is motivated by a real modeling bottleneck: single-nucleotide tokenization is expensive, while fixed k-mer and BPE schemes are brittle or biologically insensitive. The proposed architecture is reasonably coherent: the encoder learns boundaries, the main network operates on compressed segments, and the decoder reconstructs nucleotide-level representations while attempting to avoid mask-induced shortcuts. The results section is also fairly strong. On the NT benchmark, DNACHUNKER is reported as best overall average MCC and average rank, and on the Genomic Benchmark it is near the strongest baseline while using far fewer parameters than GENERator.

Soundness

I think the core idea is good, but the paper currently leaves some important technical questions unresolved. First, the explanation of masked residual gating is not yet fully convincing. The paper argues that residuals from the encoder into masked positions create a shortcut because the encoder is bidirectional and mixes neighboring context, so masked-token reconstruction could bypass the main network. That concern is directionally sensible, but it is not sufficiently differentiated from standard MLM encoders, which are also bidirectional and also reconstruct masked tokens from contextualized neighbor representations.

The paper does not show what happens if residuals into masked segments are not gated, nor whether the main-network bottleneck meaningfully matters for downstream performance, MLM loss, or the biological structure of learned segments. That omission is especially noticeable because the paper’s methodological claims lean heavily on this design choice. More broadly, Section 4.2 is labeled “Ablative Studies,” but the content is mostly additional benchmarking against other adaptive/tokenization methods plus qualitative or analytic characterization of the learned tokenizer. Those are useful experiments, but they are not ablations of the proposed system. The paper does not isolate the contribution of the two-stage chunker, the mask-protection rule, the residual-gating rule, the ratio loss, or the choice of BiMamba encoder/decoder.

A particularly important missing experiment is comparison against the same main network trained with fixed BPE and k-mer tokenizations. The current comparisons mix tokenization and architecture changes. Since the main claim is that adaptive tokenization helps, the cleanest baseline would keep the downstream architecture and pretraining recipe fixed while swapping only the tokenizer. Without that, it remains hard to quantify how much of the gain comes from learnable segmentation versus the rest of the hierarchical design.

Presentation

The paper is generally readable and the high-level story is easy to follow. In the Figure 3, the scale makes the distribution harder to interpret; a log-scaled y-axis would likely make the cross-category differences much easier to read, especially since repeat-region segment lengths extend much farther than the other categories.

Significance

I view the topic as  interesting. Tokenization is an underexamined but consequential design choice in genomic foundation models, and an adaptive compression scheme that preserves biologically salient resolution could matter in practice. The reported NT benchmark gains are impressive, especially given the parameter efficiency claims. That said, the significance is somewhat reduced by the lack of controlled ablations: if the gains cannot be attributed cleanly to adaptive tokenization itself.

Originality

The work is not proposing adaptive chunking from scratch, and the paper is transparent that related ideas exist in autoregressive sequence modeling and that MXDNA/PatchDNA also target genomic tokenization. The novel aspect is the adaptation of learnable dynamic chunking to a non-autoregressive masked DNA language model with explicit handling of MLM-specific masking artifacts. That is a credible originality claim, though again it would be stronger with ablations demonstrating that the MLM-specific design choices are necessary rather than merely reasonable.

---

> ### Author Rebuttal · Authors · 2026-03-31
>
> We thank the reviewer for their thorough and constructive feedback. We are glad the reviewer recognizes that adaptive tokenization is "a worthwhile direction" addressing "a real modeling bottleneck," and that our architecture is "coherent." We also appreciate the reviewer's acknowledgment upon performance - "impressive, especially given the parameter efficiency claims."
>
> Below we address each concern, and provide tables and figures in the following link: https://anonymous.4open.science/r/dnachunker-rebuttal-5943/reviewer_WZ97/README.md
>
> **W1: Missing controlled tokenizer ablation.**
> To address the reviewer's concerns, we train three variants: 6-mer, BPE, and our learnable tokenizer in the controlled setup. As shown in Table 1, our learnable tokenization achieves the best overall MCC (0.3902), surpassing BPE (0.3753) and 6-mer (0.3468), confirming the gains of learnable segmentation.
>
> **W2: No real ablations.**
> To address this concern, we performed an ablation study by removing or modifying each core component of DNAChunker: (1) mask protection, (2) residual gating, (3) ratio loss, and (4) the BiMamba encoder, replaced with a local CNN encoder (kernel size 7).
>
> Results are shown in Table 2. We find that all components (1-3) contribute meaningfully to the performance of DNAChunker, and among them the mask protection mechanism being the most crucial. Furthermore, utilizing a encoder with a global receptive field (*i.e.* BiMamba) rather than a local one (*i.e.* CNN), yields additional gains overall.
>
> **W3, W4: Residual gating justification is unconvincing & Q1: Encoder leakage vs standard MLM.**
> The key architectural distinction is the separation of computational roles. In a standard MLM, the encoder is the reasoning module. In DNAChunker, the encoder's sole purpose is boundary prediction for segmentation, while contextual reasoning is explicitly delegated to the main Transformer network (147.5M of 171.6M total parameters). These are architecturally distinct modules with distinct objectives.
>
> Without mask protection and residual gating, nothing prevents the encoder from learning mask-dependent shortcuts, *e.g.*, exploiting the presence of [MASK] tokens to trivially reconstruct targets without routing information through the main network. This has two consequences: (1) the encoder's capacity is wasted on reconstruction shortcuts rather than learning high-quality segmentation boundaries, and (2) any mask-dependent patterns learned in pretraining do not transfer to downstream tasks where masks are absent.
>
> We empirically validate this in Table 2: removing either one of mask protection or residual gating degrades performance by 0.05 MCC on average, confirming that both mechanims are essential for forcing the main network to serve as the primary locus of contextual reasoning.
>
> **W5: Figure 3 readability.**
> We thank the reviewer for the insightful comment. We will change the scale of the figure in the revised manuscript.
>
> **Q2: Biological claims are vague**
> In our main paper, we provided genome-wide evidence for this in Figures 1c, 2, and 3: DNAChunker systematically assigns shorter chunks to promoters, exons, and conserved regions while compressing repeats with longer chunks, whereas BPE produces a largely uniform token-size distribution across all annotation categories.
>
> To further sharpen the claim and address reviewer's concerns, we provide a motif-level fragmentation analysis. We sampled 10,000 sequences of 8,192 bp from HG38 and identified occurrences of 29 TF binding motifs (JASPAR 2024) and 3 cis-regulatory elements (CpG islands, TATA boxes, splice donor sites). The results are shown in Figure 1. For each occurrence, we counted the number of tokens it is fragmented into. DNAChunker averages 1.22 chunks per motif — keeping most motifs intact as single tokens. BPE averages 2.47, splitting nearly every motif across 2–3 tokens, and is statistically indistinguishable from a sequence-blind fixed 6-mer baseline (2.32). This provides evidence that unlike BPE, DNAChunker treats these motifs as a singular functional chunk.
>
> **Additional results**
> During the rebuttal period, we extended our evaluations to include BEND (Table 4) and DNALongBench (Table 5~10), two recently proposed benchmarks. DNAChunker achieves strong performance, attaining the best average rank on BEND while significantly outperforming Caduceus (the most recent baseline in DNALongBench) on DNALongBench. We believe these additional results further strengthen the contribution of our work.

---

> > ### Author Rebuttal · Reviewer_WZ97 · 2026-04-03
> >
> > Authors have largely addressed my concerns.

---

> > > ### Author Response · Authors · 2026-04-04
> > >
> > > We are glad that our rebuttal has addressed your concerns. We thank for the positive assessment of our work.

---

### Decision · Program_Chairs · 2026-04-30

**Decision:**

Accept (regular)

**Comment:**

The paper introduces DNAChunker, a learnable tokenization framework for DNA language models that produces adaptive, variable-length segments and demonstrates strong empirical performance along with biologically meaningful structure. Reviewers generally find the approach well-motivated and compelling, particularly after the rebuttal added controlled ablations. However, there are concerns about evaluation rigor and attribution, including earlier lack of clean tokenizer-only comparisons, potential data contamination, and limited validation beyond human-genome settings. Some reviewers also question the degree of novelty relative to prior adaptive tokenization methods and the consistency of gains across benchmarks.